# Learning Curves for Stochastic Gradient Descent on Structured Features

**Blake Bordelon & Cengiz Pehlevan**
John A. Paulson School of Engineering and Applied Sciences
Center for Brain Science
Harvard University
Cambridge, MA 02138, USA
`{blake_bordelon,cpehlevan}@g.harvard.edu`

## Abstract

The generalization performance of a machine learning algorithm such as a neural network depends in an intricate way on the structure of the data distribution. To analyze the influence of data structure on test loss dynamics, we study an exactly solveable model of stochastic gradient descent (SGD) which predicts test loss when training on features with arbitrary covariance structure. We solve the theory exactly for both Gaussian features and arbitrary features and we show that the simpler Gaussian model accurately predicts test loss of nonlinear random-feature models and deep neural networks trained with SGD on real datasets such as MNIST and CIFAR-10. We show that the optimal batch size at a fixed compute budget is typically small and depends on the feature correlation structure, demonstrating the computational benefits of SGD with small batch sizes. Lastly, we extend our theory to the more usual setting of stochastic gradient descent on a fixed subsampled training set, showing that both training and test error can be accurately predicted in our framework on real data.

## 1 Introduction

Understanding the dynamics of SGD on realistic learning problems is fundamental to learning theory. Due to the challenge of modeling the structure of realistic data, theoretical studies of generalization often attempt to derive data-agnostic generalization bounds or study the typical performance of the algorithm on high-dimensional, factorized data distributions (Engel & Van den Broeck, 2001). Realistic datasets, however, often lie on low dimensional structures embedded in high dimensional ambient spaces (Pope et al., 2021). For example, MNIST and CIFAR-10 lie on surfaces with intrinsic dimension of $\sim 14$ and $\sim 35$ respectively (Spigler et al., 2020). To understand the average-case performance of SGD in more realistic learning problems and its dependence on data, model and hyperparameters, incorporating structural information about the learning problem is necessary.

In this paper, we calculate the average case performance of SGD on models of the form $f(\mathbf{x}) = \mathbf{w} \cdot \boldsymbol{\psi}(\mathbf{x})$ for nonlinear feature map $\boldsymbol{\psi}$ trained with MSE loss. We express test loss dynamics in terms of the induced second and fourth moments of $\boldsymbol{\psi}$. Under a regularity condition on the fourth moments, we show that the test error can be accurately predicted in terms of second moments alone. We demonstrate the accuracy of our theory on random feature models and wide neural networks trained on MNIST and CIFAR-10 and accurately predict test loss scalings on these datasets. We explore in detail the effect of minibatch size, $m$, on learning dynamics. By varying $m$, we can interpolate our theory between single sample SGD ($m = 1$) and gradient descent on the population loss ($m \to \infty$). To explore the computational advantages SGD compared to standard full batch gradient descent we analyze the loss achieved at a fixed compute budget $C = tm$ for different minibatch size $m$ and number of steps $t$, trading off the number of parameter update steps for denoising through averaging. We show that generally, the optimal batch size is small, with the precise optimum dependent on the learning rate and structure of the features. Overall, our theory shows how learning rate, minibatch size and data structure interact with the structure of the learning problem to determine generalization dynamics. It provides a predictive account of training dynamics in wide neural networks.

## 1.1 OUR CONTRIBUTIONS

The novel contributions of this work are described below.

- We calculate the exact expected test error for SGD on MSE loss for arbitrary feature structure in terms of second and fourth moments as we discuss in Section 4.2. We show how structured gradient noise induced by sampling alters the loss curve compared to vanilla GD.

- For Gaussian features (or those with regular fourth moments), we compute the test error in Section 1. This theory is shown to be accurate in experiments with random feature models and wide networks in the kernel regime trained on MNIST and CIFAR-10.

- We show that for fixed compute/sample budgets and structured features with power law spectral decays, optimal batch sizes are small. We study how optimal batch size depends on the structure of the feature correlations and learning rate.

- We extend our exact theory to study multi-pass SGD on a fixed finite training set. Both test and training error can be accurately predicted for random feature models on MNIST.

## 2 RELATED WORK

The analysis of stochastic gradient descent has a long history dating back to seminal works of Polyak & Juditsky (1992) and Ruppert (1988), who analyzed time-averaged iterates in a noisy problem. Many more works have examined a similar setting, identifying how averaged and accelerated versions of SGD perform asymptotically when the target function is noisy (not a deterministic function of the input) (Flammarion & Bach, 2015; 2017; Shapiro, 1989; Robbins & Monro, 1951; Chung, 1954; Duchi & Ruan, 2021; Yu et al., 2020; Anastasiou et al., 2019; Gurbuzbalaban et al., 2021).

Recent studies have also analyzed the asymptotics of noise-free MSE problems with arbitrary feature structure to see what stochasticity arises from sampling. Prior works have found exponential loss curves problems as an upper bound (Jain et al., 2018) or as typical case behavior for SGD on unstructured data (Werfel et al., 2004). A series of more recent works have considered the over-parameterized (possibly infinite dimension) setting for SGD, deriving power law test loss curves emerge with exponents which are better than the $O(t^{-1})$ rates which arise in the noisy problem (Berthier et al., 2020; Pillaud-Vivien et al., 2018; Dieuleveut et al., 2016; Varre et al., 2021; Dieuleveut & Bach, 2016; Ying & Pontil, 2008; Fischer & Steinwart, 2020; Zou et al., 2021). These works provide bounds of the form $O(t^{-\beta})$ for exponents $\beta$ which depend on the task and feature distribution.

Several works have analyzed average case online learning in shallow and two-layer neural networks. Classical works often analyzed unstructured data (Heskes & Kappen, 1991; Biehl & Riegler, 1994; Mace & Coolen, 1998; Saad & Solla, 1999; LeCun et al., 1991; Goldt et al., 2019), but recently the hidden manifold model enabled characterization of learning dynamics in continuous time when trained on structured data, providing an equivalence with a Gaussian covariates model (Goldt et al., 2020; 2021). In the continuous time limit considered in these works, SGD converges to gradient flow on the population loss, where fluctuations due to sampling disappear and order parameters obey deterministic dynamics. Other recent works, however, have provided dynamical mean field frameworks which allow for fluctuations due to random sampling of data during a continuous time limit of SGD, though only on simple generative data models (Mignacco et al., 2020; 2021).

Studies of fully trained linear (in trainable parameters) models also reveal striking dependence on data and feature structure. Analysis for models trained on MSE (Bartlett et al., 2020; Tsigler & Bartlett, 2020; Bordelon et al., 2020; Canatar et al., 2020), hinge loss (Chatterji & Long, 2021; Cao et al., 2021; Cao & Gu, 2019) and general convex loss functions (Loureiro et al., 2021) have now been performed, demonstrating the importance of data structure for offline generalization.

Other works have studied the computational advantages of SGD at different batch sizes $m$. Ma et al. (2018) study the tradeoff between taking many steps of SGD at small $m$ and taking a small number of steps at large $m$. After a critical $m$, they observe a saturation effect where increasing $m$ provides diminishing returns. Zhang et al. (2019) explore how this critical batch size depends on SGD and momentum hyperparameters in a noisy quadratic model. Since they stipulate constant gradient noise

induced by sampling, their analysis results in steady state error rather than convergence at late times, which may not reflect the true noise structure induced by sampling.

## 3 PROBLEM DEFINITION AND SETUP

We study stochastic gradient descent on a linear model with parameters $\mathbf{w}$ and feature map $\boldsymbol{\psi}(\mathbf{x}) \in \mathbb{R}^N$ (with $N$ possibly infinite). Some interesting examples of linear models are random feature models, where $\boldsymbol{\psi}(\mathbf{x}) = \phi(\boldsymbol{G}\mathbf{x})$ for random matrix $\boldsymbol{G}$ and point-wise nonlinearity $\phi$ (Rahimi & Recht, 2008; Mei & Montanari, 2020). Another interesting linearized setting is wide neural networks with neural tangent kernel (NTK) parameterization (Jacot et al., 2020; Lee et al., 2020). Here the features are parameter gradients of the neural network function $\boldsymbol{\psi}(\mathbf{x}) = \nabla_{\boldsymbol{\theta}} f(\mathbf{x}, \boldsymbol{\theta})|_{\boldsymbol{\theta}_0}$ at initialization. We will study both of these special cases in experiments.

We optimize the set of parameters $\mathbf{w}$ by SGD to minimize a population loss of the form

$$L(\mathbf{w}) = \left\langle (\mathbf{w} \cdot \boldsymbol{\psi}(\mathbf{x}) - y(\mathbf{x}))^2 \right\rangle_{\mathbf{x} \sim p(\mathbf{x})}, \tag{1}$$

where $\mathbf{x}$ are input data vectors associated with a probability distribution $p(\mathbf{x})$, $\boldsymbol{\psi}(\mathbf{x})$ is a nonlinear feature map and $y(\mathbf{x})$ is a target function which we can evaluate on training samples. We assume that the target function is square integrable $\left\langle y(\mathbf{x})^2 \right\rangle_{\mathbf{x}} < \infty$ over $p(\mathbf{x})$. Our aim is to elucidate how this population loss evolves during stochastic gradient descent on $\mathbf{w}$. We derive a formula in terms of the eigendecomposition of the feature correlation matrix and the target function

$$\boldsymbol{\Sigma} = \left\langle \boldsymbol{\psi}(\mathbf{x})\boldsymbol{\psi}(\mathbf{x})^\top \right\rangle_{\mathbf{x}} = \sum_{k=1}^N \lambda_k \mathbf{u}_k \mathbf{u}_k^\top, \quad y(\mathbf{x}) = \sum_k v_k \mathbf{u}_k^\top \boldsymbol{\psi}(\mathbf{x}) + y_\perp(\mathbf{x}), \tag{2}$$

where $\langle y_\perp(\mathbf{x})\boldsymbol{\psi}(\mathbf{x}) \rangle = 0$. We justify this decomposition of $y(\mathbf{x})$ in the Appendix A using an eigendecomposition and show that it is general for target functions and features with finite variance.

During learning, parameters $\mathbf{w}$ are updated to estimate a target function $y$ which, as discussed above, can generally be expressed as a linear combination of features $y = \mathbf{w}^* \cdot \boldsymbol{\psi} + y_\perp$. At each time step $t$, the weights are updated by taking a stochastic gradient step on a fresh mini-batch of $m$ examples

$$\mathbf{w}_{t+1} = \mathbf{w}_t - \frac{\eta}{m} \sum_{\mu=1}^m \boldsymbol{\psi}_{t,\mu} \left( \mathbf{w}_t \cdot \boldsymbol{\psi}_{t,\mu} - y_{t,\mu} \right), \tag{3}$$

where each of the vectors $\boldsymbol{\psi}_{t,\mu}$ are sampled independently and $y_{t,\mu} = \mathbf{w}^* \cdot \boldsymbol{\psi}_{t,\mu}$. The learning rate $\eta$ controls the gradient descent step size while the batch size $m$ gives a empirical estimate of the gradient at timestep $t$. At each timestep, the test-loss, or generalization error, has the form

$$L_t = \left\langle (\mathbf{w}_t \cdot \boldsymbol{\psi}(\mathbf{x}) - \mathbf{w}^* \cdot \boldsymbol{\psi}(\mathbf{x}) - y_\perp(\mathbf{x}))^2 \right\rangle_{\mathbf{x}} = (\mathbf{w}_t - \mathbf{w}^*)^\top \boldsymbol{\Sigma}(\mathbf{w}_t - \mathbf{w}^*) + \left\langle y_\perp(\mathbf{x})^2 \right\rangle, \tag{4}$$

which quantifies exactly the test error of the vector $\mathbf{w}_t$. Note, however, that $L_t$ is a random variable since $\mathbf{w}_t$ depends on the precise history of sampled feature vectors $\mathcal{D}_t = \{\boldsymbol{\psi}_{t,\mu}\}$. Our theory, which generalizes the recursive method of (Werfel et al., 2004) allows us to compute the *expected* test loss by averaging over all possible sequences to obtain $\langle L_t \rangle_{\mathcal{D}_t}$. Our calculated learning curves are not limited to the one-pass setting, but rather can accommodate sampling minibatches from a finite training set with replacement and testing on a separate test set which we address in Section 4.4.

In summary, we will develop a theory that predicts the expected test loss $\langle L_t \rangle_{\mathcal{D}_t}$ averaged over training sample sequences $\mathcal{D}_t$ in terms of the quantities $\{\lambda_k, v_k, \left\langle y_\perp(\mathbf{x})^2 \right\rangle_{\mathbf{x}}\}$. This will reveal how the structure in the data and the learning problem influence test error dynamics during SGD. This theory is a quite general analysis of linear models on square loss, analyzing the performance of linearized models on arbitrary data distributions, feature maps $\boldsymbol{\psi}$, and target functions $y(\mathbf{x})$.

## 4 ANALYTIC FORMULAE FOR LEARNING CURVES

### 4.1 LEARNABLE AND NOISE FREE PROBLEMS

Before studying the general case, we first analyze the setting where the target function is *learnable*, meaning that there exist weights $\mathbf{w}^*$ such that $y(\mathbf{x}) = \mathbf{w}^* \cdot \boldsymbol{\psi}(\mathbf{x})$. For many cases of interest, this

is a reasonable assumption, especially when applying our theory to real datasets by fitting an atomic measure on $P$ points $\frac{1}{P}\sum_\mu \delta(\mathbf{x} - \mathbf{x}^\mu)$. We will further assume that the induced feature distribution is Gaussian so that all moments of $\psi$ can be written in terms of the covariance $\mathbf{\Sigma}$. We will remove these assumptions in later sections.

**Theorem 1.** *Suppose the features $\psi$ follow a Gaussian distribution $\psi \sim \mathcal{N}(0, \mathbf{\Sigma})$ and the target function is learnable in these features $y = \mathbf{w}^* \cdot \psi$. After $t$ steps of SGD with minibatch size $m$ and learning rate $\eta$, the expected (over possible sample sequences $\mathcal{D}_t$) test loss $\langle L_t \rangle_{\mathcal{D}_t}$ has the form*

$$\langle L_t \rangle_{\mathcal{D}_t} = \boldsymbol{\lambda}^\top \mathbf{A}^t \mathbf{v}^2 \,, \quad \mathbf{A} = (\mathbf{I} - \eta \, diag(\boldsymbol{\lambda}))^2 + \frac{\eta^2}{m} \, diag\left(\boldsymbol{\lambda}^2\right) + \frac{\eta^2}{m} \boldsymbol{\lambda}\boldsymbol{\lambda}^\top \tag{5}$$

*where $\boldsymbol{\lambda}$ is a vector containing the eigenvalues of $\mathbf{\Sigma}$ and $\mathbf{v}^2$ is a vector containing elements $(\mathbf{v}^2)_k = v_k^2 = (\mathbf{u}_k \cdot \mathbf{w}^*)^2$ for eigenvectors $\mathbf{u}_k$ of $\mathbf{\Sigma}$. The function $diag(\cdot)$ constructs a diagonal matrix with the argument vector placed along the diagonal.*

*Proof.* See Appendix B for the full derivation. We will provide a brief sketch of the proof here. The strategy of the proof relies on the fact that $\langle L_t \rangle = \text{Tr} \, \mathbf{\Sigma} \, \mathbf{C}_t$ where $\mathbf{C}_t = \left\langle (\mathbf{w}_t - \mathbf{w}^*)(\mathbf{w}_t - \mathbf{w}^*)^\top \right\rangle_{\mathcal{D}_t}$. We derive the following recursion relation for this error matrix

$$\mathbf{C}_{t+1} = (\mathbf{I} - \eta\mathbf{\Sigma})\mathbf{C}_t(\mathbf{I} - \eta\mathbf{\Sigma}) + \frac{\eta^2}{m}\left[\mathbf{\Sigma}\mathbf{C}_t\mathbf{\Sigma} + \mathbf{\Sigma}\text{Tr}\left(\mathbf{\Sigma}\mathbf{C}_t\right)\right] \tag{6}$$

The loss only depends on $c_{k,t} = \mathbf{u}_k^\top \mathbf{C}_t \mathbf{u}_k$. Solving the recurrence, $\boldsymbol{c}_t = \mathbf{A}^t \mathbf{v}^2$ and using $\langle L_t \rangle = \sum_k \lambda_k \mathbf{u}_k^\top \mathbf{C}_t \mathbf{u}_k = \sum_k c_{k,t}\lambda_k = \boldsymbol{\lambda}^\top \mathbf{A}^t \mathbf{v}^2$, we obtain the desired result. $\quad\square$

Below we provide some immediate interpretations of this result.

- The matrix $\mathbf{A}$ contains two components; a matrix $(\mathbf{I} - \eta \, diag(\boldsymbol{\lambda}))^2$ which represents the time-evolution of the loss under *average gradient updates*. The remaining matrix $\frac{\eta^2}{m}\left(diag(\boldsymbol{\lambda}^2) + \boldsymbol{\lambda}\boldsymbol{\lambda}^\top\right)$ arises due to fluctuations in the gradients, a consequence of the stochastic sampling process.

- The test loss obtained when training directly on the population loss can be obtained by taking the minibatch size $m \to \infty$. In this case, $\mathbf{A} \to (\mathbf{I} - \eta \, diag(\boldsymbol{\lambda}))^2$ and one obtains the population loss $L_t^{pop} = \sum_k v_k^2 \lambda_k (1 - \eta\lambda_k)^{2t}$. This population loss can also be obtained by considering small learning rates, i.e. the $\eta \to 0$ limit, where $\mathbf{A} = (\mathbf{I} - \eta \, diag(\boldsymbol{\lambda}))^2 + O(\eta^2)$.

- For general $\boldsymbol{\lambda}$ and $\eta^2/m > 0$, $\mathbf{A}$ is non-diagonal, indicating that the components $\{\mathbf{u}_1, ..., \mathbf{u}_k\}$ are not learned independently as $t$ increases like for $L_t^{pop}$, but rather interact during learning due to non-trivial coupling across eigenmodes at large $\eta^2/m$. This is unlike offline theory for learning in feature spaces (kernel regression), (Bordelon et al., 2020; Canatar et al., 2020), this observation of mixing across covariance eigenspaces agrees with a recent analysis of SGD, which introduced recursively defined "mixing terms" that couple each mode's evolution (Varre et al., 2021).

- Though increasing $m$ always improves generalization at fixed time $t$ (proof given in Appendix D), learning with a fixed compute budget (number of gradient evaluations) $C = tm$, can favor smaller batch sizes. We provide an example of this in the next sections and Figure 1 (d)-(f).

- The lower bound $\langle L_t \rangle \geq \boldsymbol{\lambda}^\top \boldsymbol{v}^2 \left[(1-\eta)^2 + \frac{\eta^2}{m}|\boldsymbol{\lambda}|^2\right]^t$ can be used to find necessary stability conditions on $m, \eta$. This bound implies that $\langle L_t \rangle$ will diverge if $m < \frac{\eta}{2-\eta}|\boldsymbol{\lambda}|^2$. The learning rate must be sufficiently small and the batch size sufficiently large to guarantee convergence. This stability condition depends on the features through $|\boldsymbol{\lambda}|^2 = \sum_k \lambda_k^2$. One can derive heuristic optimal batch sizes and optimal learning rates through this lower bound. See Figure 2 and Appendix C.

### 4.1.1 SPECIAL CASE 1: UNSTRUCTURED ISOTROPIC FEATURES

This special case was previously analyzed by Werfel et al. (2004) which takes $\mathbf{\Sigma} = \mathbf{I} \in \mathbb{R}^{N \times N}$ and $m = 1$. We extend their result for arbitrary $m$, giving the following learning curve

$$\langle L_t \rangle_{\mathcal{D}_t} = \left((1-\eta)^2 + \frac{1+N}{m}\eta^2\right)^t \|\mathbf{w}^*\|^2 \,, \quad \langle L_t^* \rangle_{\mathcal{D}_t} = \left(1 - \frac{m}{m+N+1}\right)^t \|\mathbf{w}^*\|^2, \tag{7}$$

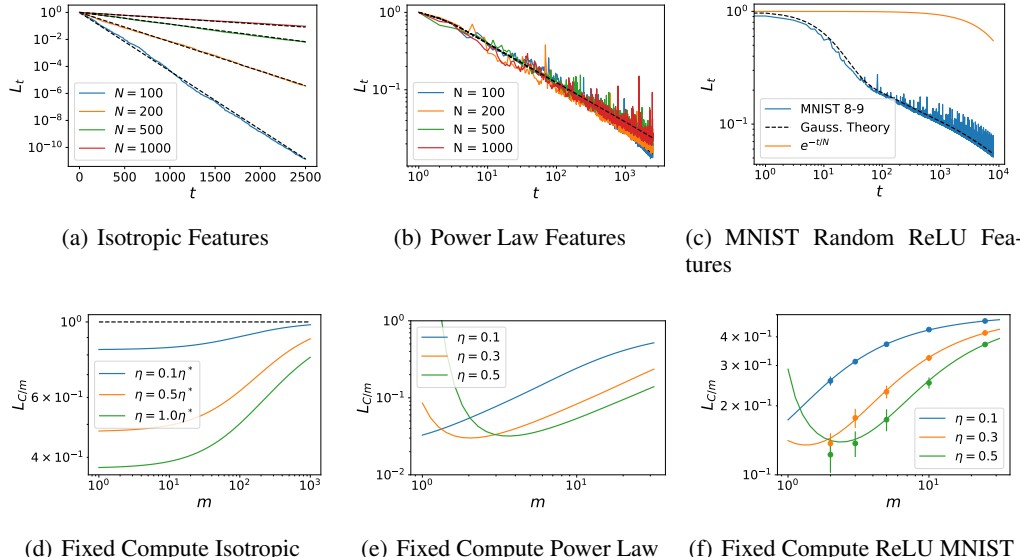

Figure 1: Isotropic features generated as $\psi \sim \mathcal{N}(0, \mathbf{I})$ have qualitatively different learning curves than power-law features observed in real data. Black dashed lines are theory. (a) Online learning with $N$-dimensional isotropic features gives a test loss which scales like $L_t \sim e^{-t/N}$ for *any target function*, indicating that learning requires $t \sim N$ steps of SGD, using the optimal learning rates $\eta^* = \frac{m}{N+m+1}$. (b) Power-law features $\psi \sim \mathcal{N}(0, \mathbf{\Lambda})$ with $\Lambda_{kl} = \delta_{k,l} k^{-2}$ have non-extensive give a *power-law scaling* $L_t \sim t^{-\beta}$ with exponent $\beta = O_N(1)$. (c) Learning to discrimninate MNIST 8's and 9's with $N = 4000$ dimensional random ReLU features (Rahimi & Recht, 2008), generates a power law scaling at large $t$, which is both quantitatively and qualitatively different than the scaling predicted by isotropic features $e^{-t/N}$. (d)-(f) The loss at a fixed compute budget $C = tm = 100$ for (d) isotropic features, (e) power law features and (f) MNIST ReLU random features with simulations (dots average and standard deviation for 30 runs). Intermediate batch sizes are preferable on real data.

where the second expression has optimal $\eta$. First, we note the strong dependence on the ambient dimension $N$: as $N \gg m$, learning happens at a rate $\langle L_t \rangle \sim e^{-tm/N}$. Increasing the minibatch size $m$ improves the exponential rate by reducing the gradient noise variance. Second, we note that this feature model has the same rate of convergence for every learnable target function $y$. At small $m$, the convergence at any learning rate $\eta$ is much slower than the convergence of the $m \to \infty$ limit, $L_{pop} = (1-\eta)^{2t} ||\mathbf{w}^*||^2$ which does not suffer from a dimensionality dependence due to gradient noise. Lastly, for a fixed compute budget $C = tm$, the optimal batch size is $m^* = 1$; see Figure 1 (d). This can be shown by differentiating $\langle L_{C/m} \rangle$ with respect to $m$ (see Appendix E). In Figure 1 (a) we show theoretical and simulated learning curves for this model for varying values of $N$ at the optimal learning rate and in Figure 1 (d), we show the loss as a function of minibatch size for a fixed compute budget $C = tm = 100$. While fixed $C$ represents fixed sample complexity, we stress that it may not represent wall-clock run time when data parallelism is available (Shallue et al., 2018).

### 4.1.2 SPECIAL CASE 2: POWER LAWS AND EFFECTIVE DIMENSIONALITY

Realistic datasets such as natural images or audio tend to exhibit nontrivial correlation structure, which often results in power-law spectra when the data is projected into a feature space, such as a randomly intialized neural network (Spigler et al., 2020; Canatar et al., 2020; Bahri et al., 2021). In the $\frac{\eta^2}{m} \ll 1$ limit, if the feature spectra and task specra follow power laws, $\lambda_k \sim k^{-b}$ and $\lambda_k v_k^2 \sim k^{-a}$ with $a, b > 1$, then Theorem 1 implies that generalization error also falls with a power law: $\langle L_t \rangle \sim Ct^{-\beta}, \quad \beta = \frac{a-1}{b}$ where $C$ is a constant. See Appendix G for a derivation with saddle point integration. Notably, these predicted exponents we recovered as a special case of our theory agree with prior work on SGD with power law spectra, which give exponents in terms of the

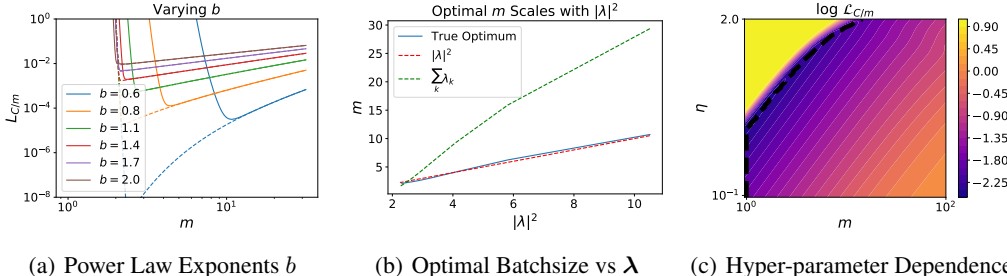

(a) Power Law Exponents $b$     (b) Optimal Batchsize vs $\boldsymbol{\lambda}$     (c) Hyper-parameter Dependence

Figure 2: Optimal batch size depends on feature structure and noise level. (a) For power law features $\lambda_k \sim k^{-b}$, $\lambda_k v_k^2 \sim k^{-a}$, the $m$ dependence of the loss $L_{C/m}$ depends strongly on the feature exponent $b$. Each color is a different $b$ value, evenly spaced in $[0.6, 2.5]$ with $a = 2.5, C = 500$. Solid lines show exact theory while dashed lines show the error predicted by approximating the mode coupling term $\frac{\eta^2}{m} \boldsymbol{\lambda}\boldsymbol{\lambda}^\top$ with decoupled $\frac{\eta^2}{m}\text{diag}(\boldsymbol{\lambda}^2)$. Mode coupling is thus necessary to accurately predict optimal $m$. (b) The optimal $m$ scales proportionally with $|\boldsymbol{\lambda}|^2 \approx \frac{1}{2b-1}$. We plot the lower bound $m_{min}$ (black), the heuristic optimum ($m$ which optimizes a lower bound for $L$, green) and $\frac{2\eta}{2-\eta}|\boldsymbol{\lambda}|^2$ (red). (c) The loss at fixed compute $C = 150$, $a = 2$, $b = 0.85$, optimal batchsize $m$ for each $\eta$ shown in dashed black. For sufficiently small $\eta$, the optimal batchsize is $m = 1$. For large $\eta$, it is better to trade off update steps for denoised gradients resulting in $m^* > 1$.

feature correlation structure (Berthier et al., 2020; Dieuleveut et al., 2016; Velikanov & Yarotsky, 2021; Varre et al., 2021). Further, our power law scaling appears to accurately match the qualitative behavior of wide neural networks trained on realistic data (Hestness et al., 2017; Bahri et al., 2021), which we study in Section 5.

We show an example of such a power law scaling with synthetic features in Figure 1 (b). Since the total variance approaches a finite value as $N \to \infty$, the learning curves are relatively insensitive to $N$, and are rather sensitive to the eigenspectrum through terms like $|\boldsymbol{\lambda}|^2$ and $\mathbf{1}^\top \boldsymbol{\lambda}$, etc. In Figure 1 (c), we see that the scaling of the loss is more similar to the power law setting than the isotropic features setting in a random features model of MNIST, agreeing excellently with our theory. For this model, we find that there can exist optimal batch sizes when the compute budget $C = tm$ is fixed (Figure 1 (e) and (f)). In Appendix C.1, we heuristically argue that the optimal batch size for power law features should scale as, $m^* \approx \frac{1}{(2b-1)}$. Figure 2 tests this result.

We provide further evidence of the existence of power law structure on realistic data in Figure 3 (a)-(c), where we provide spectra and test loss learning curves for MNIST and CIFAR-10 on ReLU random features. The eigenvalues $\lambda_k \sim k^{-b}$ and the task power tail sums $\sum_{n=k}^{\infty} \lambda_n v_n^2 \sim k^{-a+1}$ both follow power laws, generating power law test loss curves. These learning curves are contrasted with isotropically distributed data in $\mathbb{R}^{784}$ passed through the same ReLU random feature model and we see that structured data distributions allow much faster learning than the unstructured data. Our theory is predictive across variations in learning rate, batch size and noise (Figure 3).

## 4.2 ARBITRARY INDUCED FEATURE DISTRIBUTIONS: THE GENERAL SOLUTION

The result in the previous section was proven exactly for Gaussian vectors (see Appendix B). For arbitrary distributions, we obtain a slightly more involved result (see Appendix F).

**Theorem 2.** *Let $\boldsymbol{\psi}(\mathbf{x}) \in \mathbb{R}^N$ be an arbitrary feature map with covariance matrix $\boldsymbol{\Sigma} = \sum_k \lambda_k \mathbf{u}_k \mathbf{u}_k^\top$. After diagonalizing the features $\phi_k(\mathbf{x}) = \mathbf{u}_k^\top \boldsymbol{\psi}(\mathbf{x})$, introduce the fourth moment tensor $\kappa_{ijkl}^4 = \langle \phi_i \phi_j \phi_k \phi_l \rangle$. The expected loss is exactly $\langle L_t \rangle = \sum_k \lambda_k c_k(\boldsymbol{\lambda}, \boldsymbol{\kappa}, \mathbf{v}, t)$.*

We provide an exact formula for $c_k$ in the Appendix F We see that the test loss dynamics depends *only* on the second and fourth moments of the features through quantities $\lambda_k$ and $\kappa_{ijk\ell}$ respectively. We recover the Gaussian result as a special case when $\kappa_{ijkl}$ is a simple weighted sum of these three products of Kronecker tensors $\kappa_{ijkl}^{Gauss} = \lambda_i \lambda_j \delta_{ik}\delta_{jl} + \lambda_i \lambda_k \delta_{ij}\delta_{kl} + \lambda_i \lambda_j \delta_{il}\delta_{jk}$. As an alternative to

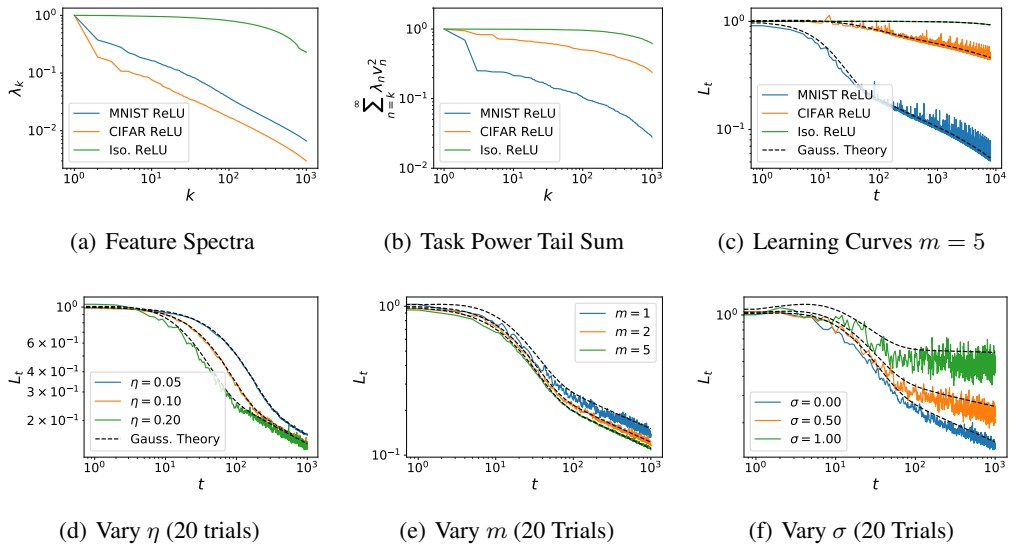

(a) Feature Spectra  (b) Task Power Tail Sum  (c) Learning Curves $m = 5$

(d) Vary $\eta$ (20 trials)  (e) Vary $m$ (20 Trials)  (f) Vary $\sigma$ (20 Trials)

Figure 3: Structure in the data distribution, nonlinearity, batchsize and learning rate all influence learning curves. (a) ReLU random feature embedding in $N = 4000$ dimensions of MNIST and CIFAR images have very different eigenvalue scalings than spherically isotropic vectors in $784$ dimensions. (b) The task power spectrum decays much faster for MNIST than for random isotropic vectors. (c) Learning curves reveal the data-structure dependence of test error dynamics. Dashed lines are theory curves derived from equation. (d) Increasing the learning rate increases the initial speed of learning but induces large fluctuations in the loss and can be worse at large $t$. Experiment curves averaged over 20 random trajectories of SGD. (e) Increasing the batch size alters both the average test loss $L_t$ and the variance. (f) Noise in the target values during training produces an asymptotic error $L_\infty$ which persists even as $t \to \infty$.

the above closed form expression for $\langle L_t \rangle$, a recursive formula which tracks $N$ mixing coefficients has also been used to analyze the test loss dynamics for arbitrary distributions (Varre et al., 2021).

Next we show that a regularity condition, similar to those assumed in other recent works (Jain et al., 2018; Berthier et al., 2020; Varre et al., 2021), on the fourth moment structure of the features allows derivation of an upper bound which is qualitatively similar to the Gaussian theory.

**Theorem 3.** *If the fourth moments satisfy* $\langle \boldsymbol{\psi}\boldsymbol{\psi}^\top \boldsymbol{G}\boldsymbol{\psi}\boldsymbol{\psi}^\top \rangle \preceq (\alpha + 1)\boldsymbol{\Sigma}\boldsymbol{G}\boldsymbol{\Sigma} + \alpha\boldsymbol{\Sigma}Tr\boldsymbol{\Sigma}\boldsymbol{G}$ *for any positive-semidefinite* $\boldsymbol{G}$, *then*

$$L_t \leq \boldsymbol{\lambda}^\top \mathbf{A}^t \mathbf{v}^2 \,, \ \mathbf{A} = (\mathbf{I} - \eta \, diag(\boldsymbol{\lambda}))^2 + \frac{\alpha\eta^2}{m}\left[diag(\boldsymbol{\lambda}^2) + \boldsymbol{\lambda}\boldsymbol{\lambda}^\top\right]. \tag{8}$$

We provide this proof in Appendix F.1. We note that the assumed bound on the fourth moments is tight for Gaussian features with $\alpha = 1$, recovering our previous theory. Thus, if this condition on the fourth moments is satisfied, then the loss for the non-Gaussian features is upper bounded by the Gaussian test loss theory with the batch size effectively altered $\tilde{m} = m/\alpha$.

The question remains whether the Gaussian approximation will provide an accurate model on *realistic data*. We do not provide a proof of this conjecture, but verify its accuracy in empirical experiments on MNIST and CIFAR-10 as shown in Figure 3. In Appendix Figure F.1, we show that the fourth moment matrix for a ReLU random feature model and its projection along the eigenbasis of the feature covariance is accurately approximated by the equivalent Gaussian model.

### 4.3 Unlearnable or Noise Corrupted Problems

In general, the target function $y(\mathbf{x})$ may depend on features which cannot be expressed as linear combinations of features $\boldsymbol{\psi}(\mathbf{x})$, $y(\mathbf{x}) = \mathbf{w}^* \cdot \boldsymbol{\psi}(\mathbf{x}) + y_\perp(\mathbf{x})$. Let $\langle y_\perp(\mathbf{x})^2 \rangle_{\mathbf{x}} = \sigma^2$. Note that $y_\perp$ need not be deterministic, but can also be a stochastic process which is uncorrelated with $\boldsymbol{\psi}(\mathbf{x})$.

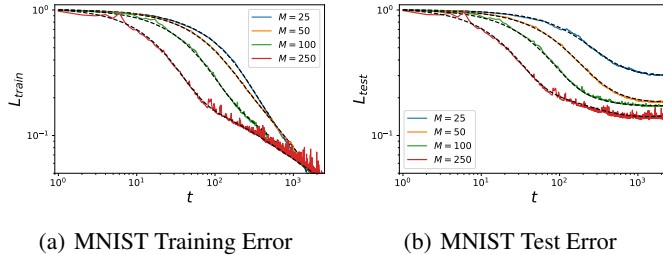

(a) MNIST Training Error         (b) MNIST Test Error

Figure 4: Training and test errors of a model trained on a training set of size $M$ can be computed with the $\mathbf{C}_t$ matrix. Dashed black lines are theory. (a) The training error for MNIST random feature model approaches zero asymptotically. (b) The test error saturates to a quantity dependent on $M$.

**Theorem 4.** *For a target function with unlearnable variance* $\langle y_\perp^2 \rangle = \sigma^2$ *trained on Gaussian* $\psi$, *the expected test loss has the form*

$$\langle L_t \rangle - \sigma^2 = \boldsymbol{\lambda}^\top \mathbf{A}^t \mathbf{v}^2 + \frac{1}{m}\eta^2\sigma^2\boldsymbol{\lambda}^\top(\mathbf{I}-\mathbf{A})^{-1}(\mathbf{I}-\mathbf{A}^t)\boldsymbol{\lambda} \tag{9}$$

*which has an asymptotic, irreducible error* $\langle L_\infty \rangle = \sigma^2 + \frac{1}{m}\eta^2\sigma^2\boldsymbol{\lambda}^\top(\mathbf{I}-\mathbf{A})^{-1}\boldsymbol{\lambda}$ *as* $t \to \infty$.

See Appendix H for the proof. The convergence to the asymptotic error takes the form $\langle L_t - L_\infty \rangle = \boldsymbol{\lambda}^\top \mathbf{A}^t \left(\mathbf{v}^2 - \frac{1}{m}\eta^2\sigma^2(\mathbf{I}-\mathbf{A})^{-1}\boldsymbol{\lambda}\right)$. We note that this quantity is not necessarily monotonic in $t$ and can exhibit local maxima for sufficiently large $\sigma^2$, as in Figure 3 (f).

### 4.4 TEST/TRAIN SPLITS

Rather than interpreting our theory as a description of the average test loss during SGD in a one-pass setting, where data points are sampled from the a distribution at each step of SGD, our theory can be suitably modified to accommodate multiple random passes over a finite training set. To accomplish this, one must first recognize that the training and test distributions are different.

**Theorem 5.** *Let* $\hat{p}(\mathbf{x}) = \frac{1}{M}\sum_\mu \delta(\mathbf{x}-\mathbf{x}^\mu)$ *be the empirical distribution on the* $M$ *training data points and let* $\hat{\boldsymbol{\Sigma}} = \langle \boldsymbol{\psi}(\mathbf{x})\boldsymbol{\psi}(\mathbf{x})^\top \rangle_{\mathbf{x}\sim\hat{p}(\mathbf{x})} = \sum_k \hat{\lambda}_k \mathbf{u}_k \mathbf{u}_k^\top$ *be the feature correlation matrix on this training set. Let* $p(\mathbf{x})$ *be the test distribution* $\boldsymbol{\Sigma}$ *its corresponding feature correlation. Then we have*

$$\langle L_{train,t} \rangle = Tr\left[\hat{\boldsymbol{\Sigma}}\mathbf{C}_t\right] , \quad \langle L_{test,t} \rangle = Tr\left[\boldsymbol{\Sigma}\mathbf{C}_t\right]$$

$$\mathbf{C}_{t+1} = (\mathbf{I}-\eta\hat{\boldsymbol{\Sigma}})\mathbf{C}_t(\mathbf{I}-\eta\hat{\boldsymbol{\Sigma}}) + \frac{\eta^2}{m}\left[\langle \boldsymbol{\psi}(\mathbf{x})\boldsymbol{\psi}(\mathbf{x})^\top \mathbf{C}_t \boldsymbol{\psi}(\mathbf{x})\boldsymbol{\psi}(\mathbf{x})^\top \rangle_{\mathbf{x}\sim\hat{p}(\mathbf{x})} - \hat{\boldsymbol{\Sigma}}\mathbf{C}_t\hat{\boldsymbol{\Sigma}}\right] \tag{10}$$

We provide the proof of this theorem in Appendix I. The interpretation of this result is that it provides the expected training and test loss if, at each step of SGD, $m$ points from the training set $\{\mathbf{x}^1, ..., \mathbf{x}^M\}$ are sampled uniformly with replacement and used to calculate a stochastic gradient. Note that while $\boldsymbol{\Sigma}$ can be full rank, the rank of $\hat{\boldsymbol{\Sigma}}$ has rank upper bounded by $M$, the number of training samples. The recurrence for $\mathbf{C}_t$ can again be more easily solved under a Gaussian approximation which we employ in Figure 4. Since learning will only occur along the $M$ dimensional subspace spanned by the data, the test error will have an irreducible component at large time, as evidenced in Figure 4. While the training errors continue to go to zero, the test errors saturate at a $M$-dependent final loss. This result can also allow one to predict errors on other test distributions.

## 5 COMPARING NEURAL NETWORK FEATURE MAPS

We can utilize our theory to compare how wide neural networks of different depths generalize when trained with SGD on a real dataset. With a certain parameterization, large width NNs are approximately linear in their parameters (Lee et al., 2020). To predict test loss dynamics with our theory, it therefore suffices to characterize the geometry of the gradient features $\boldsymbol{\psi}(\mathbf{x}) = \nabla_{\boldsymbol{\theta}} f(\mathbf{x}, \boldsymbol{\theta})$. In

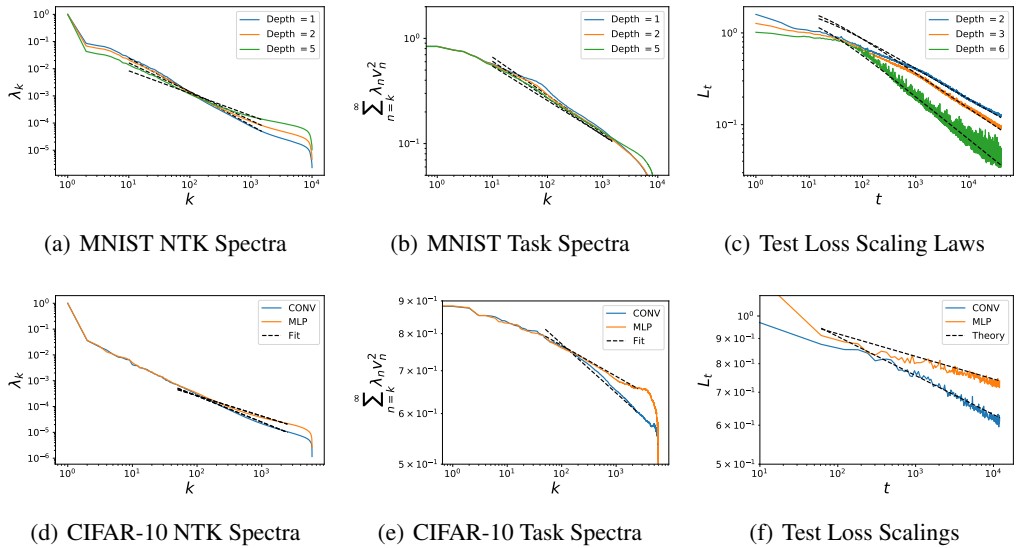

(a) MNIST NTK Spectra  (b) MNIST Task Spectra  (c) Test Loss Scaling Laws

(d) CIFAR-10 NTK Spectra  (e) CIFAR-10 Task Spectra  (f) Test Loss Scalings

Figure 5: ReLU neural networks of depth $D$ and width $500$ are trained with SGD on full MNIST. (a)-(b) Feature and spectra are estimated by diagonalizing the infinite width NTK matrix on the training data. We fit a simple power law to each of the curves $\lambda_k \sim k^{-b}$ and $v_k^2 \sim k^{-a}$. (c) Experimental test loss during SGD (color) compared to theoretical power-law scalings $t^{-\frac{a-1}{b}}$ (dashed black). Deeper networks train faster due to their slower decay in their feature eigenspectra $\lambda_k$, though they have similar task spectra. (d)-(f) The spectra and test loss for convolutional and fully connected networks on CIFAR-10. The CNN obtains a better convergence exponent due to its faster decaying task spectra. The predicted test loss scalings (dashed black) match experiments (color).

Figure 5, we show the Neural Tangent Kernel (NTK) eigenspectra and task-power spectra for fully connected neural networks of varying depth, calculated with the Neural Tangents API (Novak et al., 2020). We compute the kernel on a subset of $10,000$ randomly sampled MNIST images and estimate the power law exponents for the kernel and task spectra $\lambda_k$ and $v_k^2$. Across architectures, the task spectra $v_k^2$ are highly similar, but that the kernel eigenvalues $\lambda_k$ decay more slowly for deeper models, corresponding to a smaller exponent $b$. As a consequence, deeper neural network models train more quickly during stochastic gradient descent as we show in Figure 5 (c). After fitting power laws to the spectra $\lambda_k \sim k^{-b}$ and the task power $v_k^2 \sim k^{-a}$, we compared the true test loss dynamics (color) for a width-500 neural network model with the predicted power-law scalings $\beta = \frac{a-1}{b}$ from the fit exponents $a, b$. The predicted scalings from NTK regression accurately describe trained width-500 networks. On CIFAR-10, we compare the scalings of the CNN model and a standard MLP and find that the CNN obtains a better exponent due to its faster decaying tail sum $\sum_{n=k}^{\infty} \lambda_n v_n^2$. We stress that the exponents $\beta$ were estimated from our one-pass theory, but were utilized experiments on a finite training set. This approximate and convenient version of our theory is quite accurate across these varying models, in line with recent conjectures about early training dynamics (Nakkiran et al., 2021).

# 6  CONCLUSION

Studying a simple model of SGD, we were able to uncover how the feature geometry governs the dynamics of the test loss. We derived average learning curves $\langle L_t \rangle$ for both Gaussian and general features and showed conditions under which the Gaussian approximation is accurate. The proposed model allowed us to explore the role of the data distribution and neural network architecture on the learning curves, and choice of hyperparameters on realistic learning problems. While our theory accurately describes networks in the lazy training regime, average case learning curves in the feature learning regime would be interesting future extension. Further extensions of this work could be used to calculate the expected loss throughout curriculum learning where the data distribution evolves over time as well as alternative optimization strategies such as SGD with momentum.

## REPRODUCIBILITY STATEMENT

The code to reproduce the experimental components of this paper can be found here `https://github.com/Pehlevan-Group/sgd_structured_features`, which contains jupyter notebook files which we ran in Google Colab. More details about the experiments can be found in Appendix J. Generally, detailed derivations of our theoretical results are provided in the Appendix.

## ACKNOWLEDGEMENTS

We thank the Harvard Data Science Initiative and Harvard Dean's Competitive Fund for Promising Scholarship for their support. We also thank Jacob Zavatone-Veth for useful discussions and comments on this manuscript.

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

## A  DECOMPOSITION OF THE FEATURES AND TARGET FUNCTION

Let $y(\mathbf{x})$ be a square integrable target function with $\langle y(\mathbf{x})^2 \rangle < \infty$. Define the following integral operator $T_K$ for kernel $K(\mathbf{x}, \mathbf{x}') = \boldsymbol{\psi}(\mathbf{x}) \cdot \boldsymbol{\psi}(\mathbf{x}')$:

$$T_K[\phi](\mathbf{x}') = \int p(\mathbf{x}) K(\mathbf{x}, \mathbf{x}') \phi(\mathbf{x}) d\mathbf{x} \tag{11}$$

We are interested in eigenfunctions of this operator, function $\phi_k$ for which $T_K[\phi_k] = \lambda_k \phi_k$. For kernels with finite trace $\int K(\mathbf{x}, \mathbf{x}) p(\mathbf{x}) d\mathbf{x} < \infty$, Mercer's theorem (Rasmussen & Williams, 2005) guarantees the existence of a set of orthonormal eigenfunctions. Since $\boldsymbol{\psi}(\mathbf{x})$ spans an $N$ dimensional function space, only $N$ of the kernel eigenfunctions will have non-zero eigenvalue. Since the basis of kernel eigenfunctions (including the zero eigenvalue functions) is complete over the space of square integrable functions. After ordering the eigenvalues $\lambda_1 > \lambda_2 > ... > \lambda_N$ with $\lambda_{N+\ell} = 0$, we obtain the expansion

$$y(\mathbf{x}) = \sum_k \langle y(\mathbf{x})\phi_k(\mathbf{x}) \rangle_{\mathbf{x}} \phi_k(\mathbf{x}) = \sum_{k \leq N} v_k \phi_k(\mathbf{x}) + y_\perp(\mathbf{x}) , \ y_\perp(\mathbf{x}) = \sum_{k > N} \langle y(\mathbf{x})\phi_k(\mathbf{x}) \rangle \phi_k(\mathbf{x}) \tag{12}$$

Further, we can decompose the feature map in this basis $\boldsymbol{\psi}(\mathbf{x}) = \sum_{k=1}^N \sqrt{\lambda_k} \boldsymbol{u}_k \phi_k(\mathbf{x})$. We recognize through these decompositions the coefficients $v_k$ can be computed uniquely as $v_k = \lambda_k^{-1/2} \boldsymbol{u}_k^\top \langle \boldsymbol{\psi}(\mathbf{x}) y(\mathbf{x}) \rangle$. This provides a recipe for determining the necessary spectral quantities for our theory. We see that the feature map's decomposition above reveals that $\lambda_k$ are also the eigenvalues of the feature correlation matrix $\boldsymbol{\Sigma}$ since

$$\boldsymbol{\Sigma} = \langle \boldsymbol{\psi}(\mathbf{x})\boldsymbol{\psi}(\mathbf{x})^\top \rangle = \sum_{k\ell} \sqrt{\lambda_k \lambda_\ell} \mathbf{u}_k \mathbf{u}_\ell^\top \langle \phi_k(\mathbf{x})\phi_\ell(\mathbf{x}) \rangle = \sum_k \lambda_k \mathbf{u}_k \mathbf{u}_k^\top. \tag{13}$$

### A.1  FINITE SAMPLE SPACES

When we discuss experiments on MNIST or CIFAR, we use this technology for an atomic data distribution $p(\mathbf{x}) = \frac{1}{M} \sum_{\mu=1}^M \delta(\mathbf{x} - \mathbf{x}')$. Plugging this into the integral operator gives

$$T_K[\phi](\mathbf{x}) = \frac{1}{M} \sum_\mu K(\mathbf{x}, \mathbf{x}^\mu) \phi(\mathbf{x}^\mu) \tag{14}$$

We see that, restricting the domain to the set of points $\{\mathbf{x}_1, ..., \mathbf{x}_M\}$, this amounnts to solving a matrix eigenvalue problem $\frac{1}{M} \boldsymbol{K} \boldsymbol{\phi}_k = \lambda_k \boldsymbol{\phi}_k$ where $\boldsymbol{K} \in \mathbb{R}^{M \times M}$ is the kernel gram matrix with entries $K_{\mu\nu} = K(\mathbf{x}^\mu, \mathbf{x}^\nu)$ and $\boldsymbol{\phi}_k$ has entries $\phi_{k,\mu} = \phi_k(\mathbf{x}^\mu)$.

## B  PROOF OF THEOREM 1

Let $\boldsymbol{\Delta}_t = \mathbf{w}_t - \mathbf{w}^*$ represent the difference between the current and optimal weights and define the correlation matrix for this difference

$$\mathbf{C}_t = \left\langle \boldsymbol{\Delta}_t \boldsymbol{\Delta}_t^\top \right\rangle_{\mathcal{D}_{t-1}}. \tag{15}$$

Using stochastic gradient descent, $\mathbf{w}_{t+1} = \mathbf{w}_t - \eta \mathbf{g}_t$ with gradient vector $\mathbf{g}_t = \frac{1}{m} \sum_{i=1}^m \boldsymbol{\psi}_i \boldsymbol{\psi}_i^\top \boldsymbol{\Delta}_t$, the $\mathbf{C}_t$ matrix satisfies the recursion

$$\mathbf{C}_{t+1} = \left\langle (\boldsymbol{\Delta}_t - \eta \mathbf{g}_t)(\boldsymbol{\Delta}_t - \eta \mathbf{g}_t)^\top \right\rangle_{\mathcal{D}_t} = \mathbf{C}_t - \eta \left\langle \mathbf{g}_t \boldsymbol{\Delta}_t^\top \right\rangle - \eta \left\langle \boldsymbol{\Delta}_t \mathbf{g}_t^\top \right\rangle + \eta^2 \left\langle \mathbf{g}_t \mathbf{g}_t^\top \right\rangle. \tag{16}$$

First, note that since $\psi_i$ are all independently sampled at timestep $t$, we can break up the average into the fresh batch of $m$ samples and an average over $\mathcal{D}_{t-1}$

$$\langle \mathbf{g}_t \boldsymbol{\Delta}_t \rangle_{\mathcal{D}_t} = \frac{1}{m} \sum_{i=1}^m \left\langle \boldsymbol{\psi}_i \boldsymbol{\psi}_i^\top \right\rangle_{\boldsymbol{\psi}_i} \left\langle \boldsymbol{\Delta}\boldsymbol{\Delta}_t^\top \right\rangle_{\mathcal{D}_{t-1}} = \boldsymbol{\Sigma}\mathbf{C}_t. \tag{17}$$

The last term requires computation of fourth moments

$$\left\langle \mathbf{g}_t \mathbf{g}_t^\top \right\rangle = \frac{1}{m^2} \sum_{i,j} \left\langle \boldsymbol{\psi}_i \boldsymbol{\psi}_i^\top \left\langle \boldsymbol{\Delta}_t \boldsymbol{\Delta}_t^\top \right\rangle_{\mathcal{D}_{t-1}} \boldsymbol{\psi}_j \boldsymbol{\psi}_j^\top \right\rangle_{\boldsymbol{\psi}_i, \boldsymbol{\psi}_j} \tag{18}$$

$$= \frac{1}{m^2} \sum_{i,j} \left\langle \boldsymbol{\psi}_i \boldsymbol{\psi}_i^\top \mathbf{C}_t \boldsymbol{\psi}_j \boldsymbol{\psi}_j^\top \right\rangle_{\boldsymbol{\psi}_i, \boldsymbol{\psi}_j}. \tag{19}$$

First, consider the case where $i = j$. Letting $\boldsymbol{\psi} = \boldsymbol{\psi}_i$, we need to compute terms of the form

$$\sum_{k,\ell} C_{k,\ell} \left\langle \psi_j \psi_k \psi_\ell \psi_n \right\rangle. \tag{20}$$

For Gaussian random vectors, we resort to Wick-Isserlis theorem for the fourth moment

$$\left\langle \psi_j \psi_k \psi_l \psi_n \right\rangle = \left\langle \psi_j \psi_k \right\rangle \left\langle \psi_\ell \psi_n \right\rangle + \left\langle \psi_j \psi_\ell \right\rangle \left\langle \psi_k \psi_n \right\rangle + \left\langle \psi_j \psi_n \right\rangle \left\langle \psi_\ell \psi_k \right\rangle \tag{21}$$

giving

$$\left\langle \mathbf{g}_t \mathbf{g}_t^\top \right\rangle = \frac{m+1}{m} \boldsymbol{\Sigma} \mathbf{C}_t \boldsymbol{\Sigma} + \frac{1}{m} \boldsymbol{\Sigma} \operatorname{Tr}(\boldsymbol{\Sigma} \mathbf{C}_t). \tag{22}$$

This correlation structure for $\mathbf{g}_t$ implies that its covariance has the form

$$\left\langle \operatorname{Cov}_{\boldsymbol{\psi}}(\mathbf{g}_t) \right\rangle_{\mathcal{D}_t} = \frac{1}{m} \boldsymbol{\Sigma} \mathbf{C}_t \boldsymbol{\Sigma} + \frac{1}{m} \boldsymbol{\Sigma} \operatorname{Tr}(\boldsymbol{\Sigma} \mathbf{C}_t). \tag{23}$$

Using the formula for $\left\langle \mathbf{g}_t \mathbf{g}_t^\top \right\rangle$, we arrive at the following recursion relation for $\mathbf{C}_t$

$$\mathbf{C}_{t+1} = \mathbf{C}_t - \eta \mathbf{C}_t \boldsymbol{\Sigma} - \eta \boldsymbol{\Sigma} \mathbf{C}_t + \eta^2 \frac{m+1}{m} \boldsymbol{\Sigma} \mathbf{C}_t \boldsymbol{\Sigma} + \frac{1}{m} \eta^2 \boldsymbol{\Sigma} \operatorname{Tr}(\boldsymbol{\Sigma} \mathbf{C}_t). \tag{24}$$

Since we are ultimately interested in the generalization error $\langle L_t \rangle = \left\langle \boldsymbol{\Delta}_t^\top \boldsymbol{\Sigma} \boldsymbol{\Delta}_t \right\rangle = \operatorname{Tr} \boldsymbol{\Sigma} \mathbf{C}_t = \sum_k \lambda_k \mathbf{u}_k^\top \mathbf{C}_t \mathbf{u}_k$, it suffices to track the evolution of $c_{t,k} = \mathbf{u}_k^\top \mathbf{C}_t \mathbf{u}_k$

$$c_{t+1,k} = \left( 1 - 2\eta \lambda_k + \eta^2 \frac{m+1}{m} \lambda_k^2 \right) c_{t,k} + \frac{1}{m} \eta^2 \lambda_k \sum_j \lambda_j c_{t,j}. \tag{25}$$

Vectorizing this equation for $\mathbf{c}$ generates the following solution

$$\mathbf{c}_t = \mathbf{A}^t \mathbf{c}_0, \quad \mathbf{A} = \mathbf{I} - 2\eta \operatorname{diag}(\boldsymbol{\lambda}) + \frac{m+1}{m} \eta^2 \operatorname{diag}(\boldsymbol{\lambda}^2) + \frac{\eta^2}{m} \boldsymbol{\lambda} \boldsymbol{\lambda}^\top. \tag{26}$$

The coefficient $c_{0,k} = v_k^2 = \left( \mathbf{u}_k^\top \mathbf{w}^* \right)^2$. To get the generalization error, we merely compute $\langle L_t \rangle = \boldsymbol{\lambda}^\top \mathbf{a}_t = \boldsymbol{\lambda}^\top \mathbf{A}^t \mathbf{v}^2$ as desired.

## C  PROOF OF STABILITY CONDITIONS

We will first establish the following lower bound on the loss, where without loss of generality we assumed the maximum correlation is 1:

$$L_t \geq \boldsymbol{\lambda}^\top \mathbf{v}^2 \left[ (1 - \eta)^2 + \frac{\eta^2}{m} |\boldsymbol{\lambda}|^2 \right]^t. \tag{27}$$

We will then use this lower bound to provide necessary conditions on the learning rate and batch size for stability of the loss evolution. First, note that the following inequality holds elementwise

$$\mathbf{A}\boldsymbol{\lambda} = \left[ (\mathbf{I} - \eta \operatorname{diag}(\boldsymbol{\lambda}))^2 + \frac{\eta^2}{m} \operatorname{diag}(\boldsymbol{\lambda}^2) + \frac{\eta^2}{m} \boldsymbol{\lambda} \right] \boldsymbol{\lambda} \geq \left[ (1 - \eta)^2 + \frac{\eta^2}{m} |\boldsymbol{\lambda}|^2 \right] \boldsymbol{\lambda} \tag{28}$$

Repeating this inequality $t$ times gives $\mathbf{A}^t \boldsymbol{\lambda} \geq \left[ (1 - \eta)^2 + \frac{\eta^2}{m} |\boldsymbol{\lambda}|^2 \right]^t \boldsymbol{\lambda}$. Using the fact that $L_t = \boldsymbol{\lambda}^\top \mathbf{A}^t \mathbf{v}^2$ gives the desired inequality. Note that this inequality is very close to the true result for

isotropic features $\boldsymbol{\lambda} = \mathbf{1}$ which gives $L_t \propto \left[(1-\eta)^2 + \frac{\eta^2}{m}(|\boldsymbol{\lambda}|^2 + 1)\right]^t$. For anisotropic features with small learning rate, this bound becomes less tight. For the loss to converge to zero at large time, the quantity in brackets must necessarily be less than one. This implies the following necessary condition on the batchsize and learning rate

$$\eta < \frac{2m}{m + |\boldsymbol{\lambda}|^2} \iff m > m_{min} = \frac{\eta|\boldsymbol{\lambda}|^2}{2 - \eta} \tag{29}$$

where $m_{min}$ is the minimal batch size for learning rate $\eta$ and feature covariance eigenvalues $\boldsymbol{\lambda}$.

## C.1  Heuristic Batch Size and Learning Rate

We can derive heuristic optimal choices of the learning rate and batch size hyperparameters $\eta, m$ at a fixed compute budget which optimize the lower bound derived above.

### C.1.1  Fixed Learning Rate

We will first consider optimizing only the batch size at a fixed learning rate $\eta$ before discussing the optimal $m$ when $\eta$ is chosen optimally. The loss at a fixed compute budget $C = tm$ is lower bounded by

$$L_{C/m} \geq \boldsymbol{\lambda}^\top \boldsymbol{v}^2 \left[(1-\eta)^2 + \frac{\eta^2}{m}|\boldsymbol{\lambda}|^2\right]^{C/m} \tag{30}$$

For the purposes of optimization, we introduce $x = 1/m$ and consider optimizing

$$f(x) = x \ln\left[A + Bx\right] \ , \ A = (1-\eta)^2, B = \eta^2|\boldsymbol{\lambda}|^2 \tag{31}$$

The first order optimality condition $f'(x) = 0$ implies that $\ln(A + Bx) + \frac{Bx}{A + Bx} = 0$. Letting $z = A + Bx$, this is equivalent to $z \ln z + z - A = 0$. This equation has solutions for all valid $A \in (0, 1)$ giving solutions $z \in (e^{-1}, 1)$. Letting $z(\eta)$, represent the solution to $z + z \ln z - (1 - \eta)^2 = 0$, the optimal batchsize has the form

$$m^*(\eta) = \frac{B}{z(A) - A} = \frac{\eta^2|\boldsymbol{\lambda}|^2}{z(\eta) - (1 - \eta)^2} \tag{32}$$

We can gain intuition for this result by considering the limit of $\eta \to 0$ and $\eta \to 1$. First, in the $\eta \to 0$ limit, we find that $z \sim \frac{A+1}{2}$ so $m^* \sim \frac{2\eta|\boldsymbol{\lambda}|^2}{2 - \eta} = 2m_{min}$, making contact with the stability bound derived in Appendix Section C. Thus for small learning rates, this heuristic optimum suggests doubling the minimal stable batchsize for optimal convergence. At large learning rates, $\eta \sim 1$ with $A \sim 0$, we find $z \sim e^{-1}$ so $m^*(\eta) \sim e\eta^2|\boldsymbol{\lambda}|^2$. Thus for small $\eta$, we expect $m^*$ to scale linearly with $\eta$ while for large $\eta$, we expect a scaling of the form $\eta^2$. In either case, the optimal batchsize scales with feature eigenvalues through the sum of the squares $|\boldsymbol{\lambda}|^2 = \sum_k \lambda_k^2$.

### C.1.2  Heuristic Optimal Learning Rate and Batch Size

We will now examine what happens when one first optimizes loss bounmd with respect to the the learning rate at any value of $m$ and then subsequently optimizes over the batch size $m$. We can easily find the $\eta$ which minimizes the lower bound.

$$\frac{\partial}{\partial \eta}\left[(1-\eta)^2 + \frac{\eta^2|\boldsymbol{\lambda}|^2}{m}\right] = 0 \implies \eta^* = \frac{m}{m + |\boldsymbol{\lambda}|^2} \tag{33}$$

Note that this heuristic optimal learning rate is very close to the true optimum in the isotropic data setting $\eta_{true} = \frac{m}{m+N+1} \approx \frac{m}{m+N}$. Plugging this back into the loss bound, we find that at fixed compute $C = tm$, the loss scales like

$$L_{C/m} \geq \boldsymbol{\lambda}^\top \boldsymbol{v}^2 \left[\frac{|\boldsymbol{\lambda}|^2}{m + |\boldsymbol{\lambda}|^2}\right]^{C/m} \tag{34}$$

With the optimal choice of learning rate, the loss at fixed compute monotonically increases with batch size, giving an optimal batchsize of $m = 1$. This shows that if the learning rate is chosen optimally then small batch sizes give the best performance per unit of computation. This corresponds to the hyperparameter choices $(\eta, m) = \left(\frac{1}{1+|\boldsymbol{\lambda}|^2}, 1\right)$.

## D  INCREASING $m$ REDUCES THE LOSS AT FIXED $t$

We will show that for a fixed number of steps $t$, increasing the minibatch size $m$ can only decrease the expected error. To do this, we will simply show that the derivative of the expected loss with respect to $m$,

$$\frac{\partial \langle L_t \rangle}{\partial m} = \boldsymbol{\lambda}^\top \frac{\partial \mathbf{A}^t}{\partial m} \mathbf{v}^2, \tag{35}$$

is always non-positive. The derivative of the $t$-th power of $\mathbf{A}$ can be identified with the chain rule

$$\frac{\partial \mathbf{A}^t}{\partial m} = \frac{\partial \mathbf{A}}{\partial m} \mathbf{A}^{t-1} + \mathbf{A} \frac{\partial \mathbf{A}}{\partial m} \mathbf{A}^{t-2} + \mathbf{A}^2 \frac{\partial \mathbf{A}}{\partial m} \mathbf{A}^{t-3} + ... + \mathbf{A}^{t-1} \frac{\partial \mathbf{A}}{\partial m}. \tag{36}$$

Note that the matrix

$$\frac{\partial \mathbf{A}}{\partial m} = -\frac{\eta^2}{m^2} \left[ \text{diag}(\boldsymbol{\lambda}^2) + \boldsymbol{\lambda}\boldsymbol{\lambda}^\top \right] \tag{37}$$

has all non-positive entries. Thus we find that

$$\frac{\partial \langle L_t \rangle}{\partial m} = \sum_{n=0}^{t-1} \boldsymbol{\lambda}^\top \mathbf{A}^n \frac{\partial \mathbf{A}}{\partial m} \mathbf{A}^{t-n-1} \mathbf{v}^2. \tag{38}$$

Note that since all entries in $\mathbf{v}_k^2$ and $\mathbf{A}^{t-n-1}$ are non-negative, the vector $\boldsymbol{z}_n = \mathbf{A}^{t-n-1}\mathbf{v}^2$ has non-negative entries. By the same argument, the vector $\boldsymbol{q}_n = \mathbf{A}^n \boldsymbol{\lambda}$ is also non-negative in each entry. Therefore, each of the terms in $\frac{\partial \langle L_t \rangle}{\partial m}$ above must be non-positive

$$\frac{\partial \langle L_t \rangle}{\partial m} = \sum_{n=0}^{t-1} \boldsymbol{z}_n^\top \frac{\partial \mathbf{A}}{\partial m} \boldsymbol{q}_n = -\frac{\eta^2}{m} \sum_n \sum_{k,\ell} z_{n,k} \left[ \delta_{k,\ell} \lambda_k^2 + \lambda_k \lambda_\ell \right] q_{n,\ell} \leq 0. \tag{39}$$

Thus we find $\frac{\partial \langle L_t \rangle}{\partial m} \leq 0$, implying that optimal $\langle L_t \rangle$ is always obtained (possibly non-uniquely) at $m \to \infty$.

## E  INCREASING $m$ INCREASES THE LOSS AT FIXED $C = tm$ ON ISOTROPIC FEATURES

Unlike the previous section, which considered fixed $t$ and varying $m$, in this section we consider fixing the total number of samples (or gradient evaluations) which we call the compute budget $C = tm$. For a fixed compute budget $C = tm$, and unstructured $N$ dimensional Gaussian features and optimal learning rate $\eta^* = \frac{m}{m+N+1}$, we have

$$\langle L_{C/m} \rangle = \left( \frac{N+1}{m+N+1} \right)^{C/m} ||\mathbf{w}^*||^2. \tag{40}$$

Taking a derivative with respect to the batch size we get

$$\frac{\partial}{\partial m} \log \langle L_{C/m} \rangle = \frac{\partial}{\partial m} \frac{C}{m} \log \left( \frac{N+1}{m+N+1} \right)$$

$$= \frac{C}{m^2} \log \left( \frac{m+N+1}{N+1} \right) + \frac{C}{m(m+N+1)} > 0. \tag{41}$$

This exercise demonstrates that, for the isotropic features, smaller batch-sizes are preferred at a fixed compute budget $C$. This result does not hold for arbitrary spectra $\lambda_k$. In the general case, optimal minibatch sizes can exist as we show in Figure 1 (e)-(f) for power law and MNIST spectra.

## F  PROOF OF THEOREM 2

Let $\text{Vec}(\cdot)$ denote a flattening of an $N \times N$ matrix into a vector of length $N^2$ and let $\text{Mat}(\cdot)$ represent a flattening of a 4D tensor into a $N^2 \times N^2$ two-dimensional matrix. We will show that the expected loss (over $\mathcal{D}_t$) is

$$\langle L_t \rangle = \sum_k \lambda_k c_{t,kk} \ , \ \mathbf{c}_t = \left( \mathbf{G} + \frac{\eta^2}{m} \text{Mat}(\boldsymbol{\kappa}) \right)^t \text{Vec}(\mathbf{v}\mathbf{v}^\top) \in \mathbb{R}^{N^2} \tag{42}$$

where $[\mathbf{G}]_{ij,kl} = \delta_{ik}\delta_{jl}\left(1 - \eta(\lambda_i + \lambda_j) + \frac{\eta^2(m-1)}{m}\lambda_i\lambda_j\right)$ and $[\mathbf{v}]_k = \mathbf{u}_k \cdot \mathbf{w}^*$.

We rotate all of the feature vectors into the eigenbasis of the covariance, generating diagonalized features $\phi_k = \mathbf{u}_k^\top \boldsymbol{\psi}$ and introduce the following fourth moment tensor

$$\kappa_{ijkl} = \langle \phi_i \phi_j \phi_k \phi_\ell \rangle. \tag{43}$$

We redefine $\mathbf{C}_t$ in the appropriate (rotated) basis by projecting onto the eigenvectors of the covariance

$$\mathbf{C}_t = \mathbf{U}^\top \left\langle \boldsymbol{\Delta}_t \boldsymbol{\Delta}_t^\top \right\rangle \mathbf{U}, \tag{44}$$

where $\mathbf{U} = [\mathbf{u}_1, \mathbf{u}_2, ..., \mathbf{u}_N]$. With this definition, $\mathbf{C}$'s dynamics take the form

$$\mathbf{C}_{t+1} = \mathbf{C}_t - \boldsymbol{\Lambda}\mathbf{C}_t - \mathbf{C}_t\boldsymbol{\Lambda} + \frac{\eta^2(m-1)}{m}\boldsymbol{\Lambda}\mathbf{C}_t\boldsymbol{\Lambda} + \left\langle \boldsymbol{\phi}\boldsymbol{\phi}^\top \mathbf{C}_t \boldsymbol{\phi}\boldsymbol{\phi}^\top \right\rangle. \tag{45}$$

The elements of the matrix can be expressed with the fourth moment tensor

$$\sum_{k\ell} \langle \phi_i \phi_j \phi_k \phi_\ell \rangle C_{k\ell} = \sum_{k\ell} \kappa_{ijkl} C_{k,\ell}. \tag{46}$$

We thus generate the following dynamics for $C_{ij}^t$

$$C_{ij}^{t+1} = \left(1 - \eta(\lambda_i + \lambda_j) + \frac{\eta^2(m-1)}{m}\lambda_i\lambda_j\right)C_{ij}^t + \frac{\eta^2}{m}\sum_{kl}\kappa_{ijkl}C_{kl}^t. \tag{47}$$

Let $\mathbf{c}_t = \text{Vec}(\mathbf{C}_t)$, then we have

$$\mathbf{c}_{t+1} = \left(\mathbf{G}_0 + \frac{\eta^2}{m}\text{Mat}(\boldsymbol{\kappa})\right)\mathbf{c}_t \ , \ [\mathbf{G}_0]_{ij,k\ell} = \delta_{ik}\delta_{j\ell}\left[1 - \eta(\lambda_i + \lambda_j) + \frac{\eta^2(m-1)}{m}\lambda_i\lambda_j\right]. \tag{48}$$

Solving these dynamics for $\mathbf{c}$, recognizing that $\mathbf{c}_0 = \text{Vec}(\mathbf{v}\mathbf{v}^\top)$, and computing $\langle L_t \rangle = \text{Tr}\boldsymbol{\Sigma}\mathbf{C}_t = \sum_k C_{kk}\lambda_k$ gives the desired result.

## F.1 PROOF OF THEOREM 3

Suppose that the features satisfy the regularity condition

$$\left\langle \boldsymbol{\psi}\boldsymbol{\psi}^\top \mathbf{G} \boldsymbol{\psi}\boldsymbol{\psi}^\top \right\rangle \preceq (\alpha+1)\boldsymbol{\Sigma}\mathbf{G}\boldsymbol{\Sigma} + \alpha\boldsymbol{\Sigma}\text{Tr}\left(\boldsymbol{\Sigma}\mathbf{G}\right) \tag{49}$$

Recalling the recursion relation for $\mathbf{C}_t$

$$\mathbf{C}_{t+1} = \mathbf{C}_t - \eta\boldsymbol{\Sigma}\mathbf{C}_t - \eta\mathbf{C}_t\boldsymbol{\Sigma} + \eta^2\frac{m^2 - m}{m^2}\boldsymbol{\Sigma}\mathbf{C}_t\boldsymbol{\Sigma} + \frac{\eta^2}{m}\left\langle \boldsymbol{\psi}\boldsymbol{\psi}^\top \mathbf{C}_t \boldsymbol{\psi}\boldsymbol{\psi}^\top \right\rangle$$

$$\preceq \mathbf{C}_t - \eta\boldsymbol{\Sigma}\mathbf{C}_t - \eta\mathbf{C}_t\boldsymbol{\Sigma} + \eta^2\frac{m^2 - m}{m^2}\boldsymbol{\Sigma}\mathbf{C}_t\boldsymbol{\Sigma} + \frac{\eta^2}{m}\left[(\alpha+1)\boldsymbol{\Sigma}\mathbf{C}_t\boldsymbol{\Sigma} + \alpha\boldsymbol{\Sigma}\text{Tr}\mathbf{C}_t\boldsymbol{\Sigma}\right] \tag{50}$$

$$= (\mathbf{I} - \eta\boldsymbol{\Sigma})\mathbf{C}_t(\mathbf{I} - \eta\boldsymbol{\Sigma}) + \frac{\alpha\eta^2}{m}\left[\boldsymbol{\Sigma}\mathbf{C}_t\boldsymbol{\Sigma} + \boldsymbol{\Sigma}\text{Tr}\boldsymbol{\Sigma}\mathbf{C}_t\right] \tag{51}$$

Defining that $c_{k,t} = \boldsymbol{u}_k^\top \mathbf{C}_t \boldsymbol{u}_k$, we note $\mathbf{c}_{t+1} \leq \left((\boldsymbol{I} - \eta\,\text{diag}(\boldsymbol{\lambda}))^2 + \frac{\eta^2\alpha}{m}\left[\text{diag}(\boldsymbol{\lambda}^2) + \boldsymbol{\lambda}\boldsymbol{\lambda}^\top\right]\right)\mathbf{c}_t$. Using the fact that $L_t = \mathbf{c}_t^\top \boldsymbol{\lambda}$, we find

$$L_t \leq \boldsymbol{\lambda}^\top \left((\boldsymbol{I} - \eta\,\text{diag}(\boldsymbol{\lambda}))^2 + \frac{\eta^2\alpha}{m}\left[\text{diag}(\boldsymbol{\lambda}^2) + \boldsymbol{\lambda}\boldsymbol{\lambda}^\top\right]\right)^t \boldsymbol{v}^2 \tag{52}$$

which proves the desired bound.

## G POWER LAW SCALINGS IN SMALL LEARNING RATE LIMIT

By either taking a small learning rate $\eta$ or a large batch size, the test loss dynamics reduce to the test loss obtained from gradient descent on the population loss. In this section, we consider the small learning rate limit $\eta \to 0$, where the average test loss follows

$$\langle L_t \rangle \sim \sum_{k=1}^{\infty} \lambda_k v_k^2 (1 - \eta\lambda_k)^{2t}. \tag{53}$$

 

(a) Non-Gaussian Effects on MNIST            (b) Non-Gaussian Effects on CIFAR

Figure F.1: Non-Gaussian effects are small on random feature models. (a)-(b) The first 20-dimensions of the summed fourth moment matrix $\kappa_{ij}^4 = \mathbf{u}_i^\top \left\langle \psi\psi^\top \psi\psi^\top \right\rangle \mathbf{u}_j$ are plotted for the Gaussian approximation and the empirical fourth moment. Differences between the Gaussian approximation and true fourth moment matrices on this example are visible, but are only on the order of $\sim 5\%$ of the size of the entries in $\kappa_4$.

Under the assumption that the eigenvalue and target function power spectra both follow power laws $\lambda_k \sim k^{-b}$ and $v_k^2 \lambda_k \sim k^{-a}$, the loss can be approximated by an integral over all modes $k$

$$\langle L_t \rangle = \sum_k k^{-a}(1 - \eta k^{-b})^{2t} \sim \int_1^\infty \exp\left(2\eta \ln(1 - \eta k^{-b})t - a \ln k\right) dk \tag{54}$$

$$\sim \int_1^\infty \exp\left(-2\eta\eta k^{-b}t - a \ln k\right) dk \ , \ \eta \to 0 \tag{55}$$

We identify the function $f(k) = 2\eta k^{-b} + \frac{1}{t} \ln k$ and proceed with Laplace's method (Bender & Orszag, 1999). This consists of Taylor expanding $f(k)$ around its minimum to second order and computing a Gaussian integral

$$\int \exp(-tf(k))dk \sim \int \exp\left(-tf(k^*) - \frac{t}{2}f''(k^*)(k - k^*)^2\right) \sim \exp(-tf(k^*))\sqrt{\frac{2\pi}{tf''(k^*)}}. \tag{56}$$

We must identify the $k^*$ which minimizes $f(k)$. The interpretation of this value is that it indexes the mode which dominates the error at a large time $t$. The first order condition gives

$$f'(k) = -2b\eta k^{-b-1} + \frac{a}{tk} = 0 \implies k^* = (2b\eta t/a)^{1/b}. \tag{57}$$

The second derivative has the form

$$f''(k^*) = 2b^2 \eta k^{-b-2} - \frac{1}{tk^2}|_{k^*} = 2b^2 \eta \left(2b\eta t/a\right)^{-(b+2)/b} - \frac{1}{t}\left(2b\eta t/a\right)^{-2/b} \sim t^{-1-2/b}. \tag{58}$$

Thus we are left with a scaling of the form

$$\langle L_t \rangle \sim \exp(-a/b \ln t)t^{1/b} \sim t^{-\frac{a-1}{b}}. \tag{59}$$

## H  PROOF OF THEOREM 4

Let $\left\langle y_\perp^2 \right\rangle = \sigma^2$ and $\langle y_\perp \rangle = 0$, $\langle y_\perp \psi \rangle = \mathbf{0}$. The gradient descent updates take the following form $\mathbf{\Delta}_{t+1} = \mathbf{\Delta}_t - \eta\mathbf{g}_t$ with

$$\mathbf{g}_t = \frac{1}{m} \sum_{i=1}^m \psi_i \left[\psi_i^\top \mathbf{\Delta}_t + y_{\perp,i}\right]. \tag{60}$$

Again, defining $\mathbf{C}_t = \left\langle \mathbf{\Delta}_t \mathbf{\Delta}_t^\top \right\rangle$ we perform the average over each of the $\psi_i$ vectors to obtain the following recursion relation

$$\mathbf{C}_{t+1} = \left\langle \mathbf{\Delta}_t \mathbf{\Delta}_t^\top \right\rangle - \eta \left\langle \mathbf{\Delta}_t \mathbf{g}_t^\top \right\rangle - \eta \left\langle \mathbf{g}_t \mathbf{\Delta}_t^\top \right\rangle + \eta^2 \left\langle \mathbf{g}_t \mathbf{g}_t^\top \right\rangle$$

$$= \mathbf{C}_t - \eta\mathbf{\Sigma}\mathbf{C}_t - \eta\mathbf{C}_t\mathbf{\Sigma} + \frac{m+1}{m}\mathbf{\Sigma}\mathbf{C}_t\mathbf{\Sigma} + \frac{1}{m}\mathbf{\Sigma}\mathrm{Tr}\left(\mathbf{C}_t\mathbf{\Sigma}\right) + \eta^2\sigma^2\mathbf{\Sigma}. \tag{61}$$

Again, analyzing $c_{t,k} = \mathbf{u}_k^\top \mathbf{C}_t \mathbf{u}_k$ we find

$$c_{t+1,k} = \left(1 - 2\eta\lambda_k + \eta^2 \frac{m+1}{m}\lambda_k^2\right) c_{t,k} + \frac{1}{m} \sum_\ell \lambda_\ell c_{t,\ell} + \eta^2 \sigma^2 \lambda_k. \tag{62}$$

The vector $c_t$ follows the linear evolution

$$c_{t+1} = \mathbf{A}c_t + \eta^2\sigma^2\boldsymbol{\lambda}. \tag{63}$$

Let $\boldsymbol{b} = \eta^2\sigma^2\boldsymbol{\lambda}$. Writing out the first few steps, we identify a pattern

$$c_1 = \mathbf{A}c_0 + \boldsymbol{b}$$
$$c_2 = \mathbf{A}c_1 + \boldsymbol{b} = \mathbf{A}^2 c_0 + \mathbf{A}\boldsymbol{b} + \boldsymbol{b}$$
$$c_3 = \mathbf{A}c_2 + \boldsymbol{b} = \mathbf{A}^3 c_0 + \mathbf{A}^2\boldsymbol{b} + \mathbf{A}\boldsymbol{b} + \boldsymbol{b}$$
$$...$$
$$c_t = \mathbf{A}^t c_0 + \left(\sum_{n=0}^{t-1} \mathbf{A}^n\right)\boldsymbol{b}. \tag{64}$$

The geometric sum $\left(\sum_{n=0}^{t-1} \mathbf{A}^n\right)$ can be computed exactly under the assumption that $(\mathbf{I} - \mathbf{A})$ is invertible which holds provided all of $\mathbf{A}$'s eigenvalues are less than unity, which necessarily holds provided the system is stable. The geometric sum yields

$$\left(\sum_{n=0}^{t-1} \mathbf{A}^n\right) = (\mathbf{I} - \mathbf{A})^{-1}\left(\mathbf{I} - \mathbf{A}^t\right). \tag{65}$$

Recalling the definition of $\boldsymbol{b} = \sigma^2\eta^2\boldsymbol{\lambda}$ and the definition of the average loss $\langle L_t\rangle = \boldsymbol{\lambda}^\top c_t$, we have

$$\langle L_t\rangle = \sigma^2 + \boldsymbol{\lambda}^\top \mathbf{A}^t c_0 + \eta^2\sigma^2\boldsymbol{\lambda}^\top (\mathbf{I} - \mathbf{A})^{-1}\left(\mathbf{I} - \mathbf{A}^t\right)\boldsymbol{\lambda}. \tag{66}$$

Recognizing $c_0 = \mathbf{v}^2$ gives the desired result.

## I  PROOF OF THEOREM 5

We will now prove that if the training $\hat{p}(\mathbf{x})$ and test distributions $p(\mathbf{x})$ are different and have feature correlation matrices $\hat{\boldsymbol{\Sigma}}$ and $\boldsymbol{\Sigma}$ respectively, then the average training and test losses have the form

$$L_{\text{train}} = \text{Tr}\left[\hat{\boldsymbol{\Sigma}}\mathbf{C}_t\right] \quad L_{\text{test}} = \text{Tr}\left[\boldsymbol{\Sigma}\mathbf{C}_t\right]. \tag{67}$$

As before, we will assume that there exist weights $\mathbf{w}^*$ which satisfy $y = \mathbf{w}^*\cdot\psi$. We start by noticing that the update rule for gradient descent

$$\mathbf{w}_t = \mathbf{w}_t - \eta\mathbf{g}_t \ , \ \mathbf{g}_t = \frac{1}{m}\sum_{\mu=1}^{m}\boldsymbol{\psi}_{t,\mu}\boldsymbol{\psi}_{t,\mu}^\top[\mathbf{w}_t - \mathbf{w}^*] \tag{68}$$

generates the following dynamics for the weight discrepancy correlation $\mathbf{C}_t = \left\langle(\mathbf{w}_t - \mathbf{w}^*)(\mathbf{w}_t - \mathbf{w}^*)^\top\right\rangle_{\mathcal{D}_t}$.

$$\mathbf{C}_{t+1} = \mathbf{C}_t - \eta\hat{\boldsymbol{\Sigma}}\mathbf{C}_t - \eta\mathbf{C}_t\hat{\boldsymbol{\Sigma}} + \eta^2\left(1 - \frac{1}{m}\right)\hat{\boldsymbol{\Sigma}}\mathbf{C}_t\hat{\boldsymbol{\Sigma}} + \frac{\eta^2}{m}\left\langle\boldsymbol{\psi}\boldsymbol{\psi}^\top\mathbf{C}_t\boldsymbol{\psi}\boldsymbol{\psi}^\top\right\rangle \tag{69}$$

This formula can be obtained through the simple averaging procedure shown in B. Under the Gaussian approximation, we can obtain a simplification

$$\mathbf{C}_{t+1} = (\mathbf{I} - \eta\boldsymbol{\Sigma})\mathbf{C}_t(\mathbf{I} - \eta\boldsymbol{\Sigma}) + \frac{\eta^2}{m}\left[\boldsymbol{\Sigma}\mathbf{C}_t\boldsymbol{\Sigma} + \boldsymbol{\Sigma}\text{Tr}\boldsymbol{\Sigma}\mathbf{C}_t\right] \tag{70}$$

We can solve for the evolution of the diagonal and off-diagonal entries in this matrix giving

$$\mathbf{u}_k^\top\mathbf{C}_t\mathbf{u}_k = \left[\mathbf{A}^t\mathbf{v}^2\right]_k \ , \ \mathbf{u}_k^\top\mathbf{C}_t\mathbf{u}_\ell = \left(1 - \eta\hat{\lambda}_k - \eta\hat{\lambda}_\ell + \eta^2\left(1 + \frac{1}{m}\right)\hat{\lambda}_k\hat{\lambda}_\ell\right)^t v_k v_\ell \tag{71}$$

To calculate the training and test error, we have

$$\langle L_{\text{test}} \rangle = \left\langle (\boldsymbol{\psi}(\mathbf{x}) \cdot \mathbf{w}_t - \boldsymbol{\psi}(\mathbf{x}) \cdot \boldsymbol{w}^*)^2 \right\rangle_{\mathbf{x} \sim p(\mathbf{x}), \mathcal{D}_t} = \langle (\mathbf{w}_t - \mathbf{w}^*) \boldsymbol{\Sigma} (\mathbf{w}_t - \mathbf{w}^*) \rangle = \text{Tr}\boldsymbol{\Sigma}\mathbf{C}_t.$$

$$\langle L_{\text{train}} \rangle = \left\langle (\boldsymbol{\psi}(\mathbf{x}) \cdot \mathbf{w}_t - \boldsymbol{\psi}(\mathbf{x}) \cdot \boldsymbol{w}^*)^2 \right\rangle_{\mathbf{x} \sim \hat{p}(\mathbf{x}), \mathcal{D}_t} = \left\langle (\mathbf{w}_t - \mathbf{w}^*) \hat{\boldsymbol{\Sigma}} (\mathbf{w}_t - \mathbf{w}^*) \right\rangle = \text{Tr}\hat{\boldsymbol{\Sigma}}\mathbf{C}_t. \quad (72)$$

Note that in the training error formula, since $\hat{\boldsymbol{\Sigma}}$ has eigenvectors $\mathbf{u}_k$ only the diagonal terms $\mathbf{u}_k^\top \mathbf{C}_t \mathbf{u}_k$ enter into the formula for $L_{train}$, but off-diagonal components $\mathbf{u}_k^\top \mathbf{C}_t \mathbf{u}_\ell$ do enter into the formula for $L_{test}$

## J  EXPERIMENTAL DETAILS

For Figures 3, we use the last two classes of MNIST and CIFAR-10. We encode the target values as binary $y \in \{+1, -1\}$. For Figure 5, we use 6000 random training points drawn from entire MNIST and CIFAR-10 datasets to calculate the spectrum of the Fisher information matrix. We train with SGD on these training data, using one-hot label vectors for each training example and plot the error on the test set. We train our models on a Google Colab GPU and include code to reproduce all experimental results in the supplementary materials. To match our theory, we use fixed learning rate SGD. Both evaluation of the infinite width kernels and training were performed with the Neural Tangents API (Novak et al., 2020).

