# OpenReview forum: "Learning Curves for SGD on Structured Features"
_ICLR.cc/2022/Conference — ICLR 2022 Poster_

### Official Review · Reviewer_AoHd · 2021-11-02

**Correctness:** 3
**Technical Novelty And Significance:** 2
**Empirical Novelty And Significance:** 2
**Recommendation:** 5
**Confidence:** 3

**Main Review:**

Strengths:
- The paper analyzes the influence of data structure (e.g., the spectrum of covariance) on the loss dynamics, which is very interesting.

Weaknesses:
- While this paper studies an interesting problem, the contributions and novelty of the paper are unclear. Similar models have been studied extensively for either least-squares regression or neural network training. For all the subsections in section 3, there are similar works done before. I believe the presentation of the paper could be largely improved by first explicitly stating the existing work (rather than deferring the related work to the end of the paper) and then introducing the novel contributions of this work. For example, [1, 2] did very similar analyses about the learning dynamics of SGD or other algorithms. In addition, [1] also looked at power-law distribution.
- For the discussion on power-law eigenspectrum, it seems that the authors assume the dimensionality is infinite which leads to the power-law error dynamics. However, the features are often finite dimension (other than kernel methods), so the motivation of assuming infinite dimensionality is unclear to me. It is important to clarify this assumption since the authors use this model to predict neural network training dynamics. To my knowledge, the power-law error in practice is due to the label noise, rather than the artificial result of infinite dimensionality.
- The discussion on the relationship between optimal batch size and feature covariance is new. However, I don’t find it surprising as many empirical investigations have shown that the optimal batch size depends on the specific dataset.


[1] Which algorithmic choices matter at which batch sizes?
[2] Accelerating Stochastic Gradient Descent For Least Squares Regression



**Summary Of The Paper:**

The paper studies the learning dynamics of stochastic gradient descent on simple linear models with structured features. In particular, the paper discussed the influence of data structure on learning dynamics and optimal batch sizes. In practice, this model seems to be able to predict the training/test error of small neural networks on real data.


**Summary Of The Review:**

This paper studies an interesting problem, but its contributions are unclear to me as similar models have been studied before. Rederiving the loss dynamics with different data structures makes the paper a bit incremental. Overall, this is a borderline paper and I'm inclined to vote for a reject. That being said, I'm happy to increase the score if the authors could reorganize the paper and emphasize their novel contributions.

---

> ### Author Response · Authors · 2021-11-18
> **Response to AoHd**
>
> We thank the reviewer for the helpful comments and interesting question about power law scalings.
>
> 1. We extended our literature review. If there is anything we missed, we would be happy to add it to our review. While we did try connecting our general result (for arbitrary spectra) to special cases (isotropic and power law features) where approximate results or bounds were obtained in prior works, our general and exact formulas (for both Gaussian and non-Gaussian features) are novel. Further, unlike many prior works, we do not give an adhoc model of gradient noise but rather directly average the test loss over the history random samples. We frequently rely on the predictive utility of our new theory in our experiments. The spectra for random feature models on realistic data are not exactly power laws and the theory curves we plot rely on analyzing the full spectrum. We only use the predicted power law exponents in the NTK results in Figure 5.
>
> 2.  We thank the reviewer for this constructive suggestion. We reorganized the paper so that related work and our listed contributions preceed the main presentation of our result. We think this makes the paper more readable and places our results in the literature. We added a concrete list of our novel contributions.
>
> 3. Though both papers [1,2] discuss power law features, we would like to stress the difference between the setting we consider and the settings of these papers. The paper [1] does study the power law $\lambda_k = k^{-1}$, however we deal with arbitrary task and feature power laws. Rather than solving for the induced distribution of gradient noise by studying the SGD update on $m$ random samples at each timestep, the work [1] assumes that gradient noise covariance is (A) constant in time and (B) codiagonalizable with the feature correlations. Our work shows that neither assumption is generally true for arbitrary features. As a consequence, many predictions in [1] differ from our results, including their prediction that SGD on a deterministic target function will have a steady state error since gradient noise does not decrease with time. Our theory, in contrast, shows that noise variance is automatically annealed since the size of the gradient covariance decreases with time. The paper [2] studies similar regularity conditions on fourth moments is now cited when we discuss this. Many of the bounds derived in [2] involve the condition number which depends on the minimum eigenvalue of the covariance being non-vanishing.
>
> 4. In a finite dimensional setting, it is true the loss can always be upper bounded in terms of the condition number as is done in [2] or in our theory through the operator norm of the matrix $A$. However, we show that in many cases this bound is not descriptive, for example in Figure 1 (b, c). As a concrete example consider a setting where $\lambda_{max} = 1$ and $\lambda_{min} = \epsilon$ for some arbitrarily small $\epsilon$. The rate $(1-\eta\epsilon)^{2t} \sim e^{-2 \epsilon \eta t}$ will not be descriptive for times $t \ll 1/\epsilon$, but the power law scaling $t^{-\beta}$ (derived from decay rates of feature and task spectra) can be. This is the reason why the power law we derived is actually descriptive for a wide range of times for a variety of $N$ values as we show in Figure 1 (b), not just for infinite feature spaces. In the true $N\to\infty$ limit, the minimum eigenvalue tends to zero (so a bound based on condition number would become vacuous) but the power law scaling becomes exact. This is also discussed in the recent work of Varre et al 2021 (https://arxiv.org/abs/2102.03183). The power law with exponent $\beta$ is also not a consequence of label noise, but rather arises simply through the structure of the features, as we show in 1 (b) and Figure 5. In the Appendix G we show why this scaling can arise in a large $N$ situation through a saddle-point integration argument at large $t$.
>
>
>
>
> 5. That there would be a dependence of optimal batch size on features may be unsurprising, but our theory provides an exact expression for this dependence, namely that the optimal $m$ should scale with the sum of the squares of feature eigenvalues $\sum_k \lambda_k^2$. This is surprising since the total variance of features is the sum of the eigenvalues $\sum_k \lambda_k$, rather than the sum of the squares.  We added a new Figure 2 (b) comparing these scalings, showing that our proposal matches the true optimal batch size. We also add a visualization of loss at fixed compute in the $(\eta, m)$ plane in the new Figure 2 c.

---

> > ### Comment · Reviewer_AoHd · 2021-11-18
> > **Two remaining questions**
> >
> > Thank you for the rebuttal. The current organization makes the paper easier to follow.
> >
> > I still have two questions.
> >
> > - You mentioned in your response that "noise variance is automatically annealed since the size of the gradient covariance decreases with time." Isn't that only the case for training? I believe for test/validation, there's still an irreducible term if a constant learning rate is used. If I understand correctly, [1] assumes an online setting, so the additive noise (constant covariance) follows from the central limit theorem (see e.g. [3]). Perhaps, the online setting does match the learning curves on the validation/test set well in practice (see [4]).
> > - The power-law test loss curve assumes the target function is learnable. I don't think that's the case for real datasets. With finite examples for training, I believe the test loss would eventually saturate to a non-zero constant. What would happen if you train the models longer in Figure 5 (c) and (f)? Would the test errors saturate? If the target function is unlearnable, then why would one expect the empirical test loss can be predicted with the power-law curves (that assume learnable target functions)?
> >
> > Reference
> > 3. Stochastic Gradient Descent as Approximate Bayesian Inference
> > 4. The Deep Bootstrap Framework: Good Online Learners are Good Offline Generalizers

---

> > > ### Author Response · Authors · 2021-11-18
> > > **response to remaining questions**
> > >
> > > 1. For the first question, it is helpful to distinguish between the one-pass and multiple pass setting. In both cases, the gradient covariance will decrease with time (note that at convergence the gradients of the model should be zero). The reviewer is correct though that the multiple pass setting will have an irreducible error at late time since the model only has access to the M training points.
> > >
> > > 2. The power law expressions for test error were to be interpreted in the one-pass setting where there is not an irreducible loss.  In the multiple pass setting, a power law training error curves are possible. If you have $M \gg t * m$ then the difference between training in the multiple pass setting and one-pass setting should be small as discussed in the deep bootstrap paper reference 3: short time training on batched data should coincide with test error so the early time test error should match the prediction of the online theory. If we continued training on Figure 5c and passed over the data multiple times the error would saturate.
> > >
> > > If the reviewer thinks making these points explicitly would help, we would be glad to expand on it in the paper.

---

> > > > ### Comment · Reviewer_AoHd · 2021-11-18
> > > > **Further response**
> > > >
> > > > What made you think the gradient covariance would go to zero? Let's say the targets follow the function $y = w^* x + \epsilon$ with $\epsilon \sim N(0, \sigma^2)$. I believe it goes to zero only when it is learnable (I think most problems are noise corrupted in practice).
> > > >
> > > > It is better to make it clear about learnable vs unlearnable, one-pass vs multi-pass in the paper.

---

> > > > > ### Author Response · Authors · 2021-11-18
> > > > > **Further Response**
> > > > >
> > > > > Thank you for the great question!
> > > > >
> > > > > There is a difference between the irreducible error in the train/test split and the irreducible error when the target function is unlearnable over the training distribution. We write the decomposition of $y = w * x + \epsilon$ when $\epsilon$ is the component of $y$ uncorrelated with features *over the distribution used for training*; ie $\left< \epsilon(x) \psi(x) \right>=0$ where average is over the *training distribution*. This is what causes the extra noise in the gradients and is natural in the one pass setting (where $p_{train} = p_{test}$) if the function $y$ is not estimable from the features $\psi$.
> > > > >
> > > > > However, in the finite training set setting, if the features are sufficiently rich to interpolate the training set (the $M \times M$ kernel gram matrix $K_{\mu,\nu} = \psi_{\mu} \cdot \psi_{\nu}$ on the dataset is full rank), then $\sigma=0$ since the equation $K \alpha = y$ can be solved. This will generically happen in the overparameterized setting when $N>M$.
> > > > >
> > > > > Concretely, we can explicitly work out the gradient covariance at time $t$ in the finite training set setting has the form $\text{Cov}(g_t) = \Sigma_{train} C_t \Sigma_{train} + \Sigma L_{train}$.  We show that $C_t$ decays to zero along the non-zero eigendirections of the training distribution correlation $\Sigma_{train}$, and that $L_{train}$ goes to zero. Thus the gradient noise should go to zero at long times.
> > > > >
> > > > > The irreducible *test error* due to test/train splits arises from the fact that $C_t$ does not decay to zero in dimensions other than the $M$ eigendirections of $\Sigma_{train}$ which have nonzero eigenvalue. Thus $\lim_{t\to\infty} \text{Tr} C_t \Sigma_{test} \neq 0$.

---

> > > > > > ### Comment · Reviewer_AoHd · 2021-11-19
> > > > > > **Clarification**
> > > > > >
> > > > > > Thank you for the explanation between two irreducible error terms.
> > > > > >
> > > > > > If I understand correctly, the model in [1] basically assumes the target function is unlearnable, which gives rise to additive noise (this part can be assumed to be constant). Or put it differently, they basically assume the model is sort of underparameterized. In my opinion, I feel this is always the case. As suggested by the deep bootstrap paper, the test curve is basically the training curve of the online setting. In the online setting, I don't think there is any "overparameterized" model.

---

> > > > > > > ### Author Response · Authors · 2021-11-19
> > > > > > > **Follow up on clarification**
> > > > > > >
> > > > > > > We agree with the intuition that models in general will not be able to achieve zero *population loss*, even if trained in the one-pass online setting. However, we tend  they are generally "over-parameterized" in the sense that they can achieve zero *training loss* in the multi-pass setting when training on a finite train set of size $M$.  In this second world, the gradients the model sees will eventually tend to zero as described above.
> > > > > > >
> > > > > > > We would like to stress that our theory is sufficiently general to deal with both of the two cases
> > > > > > > 1. the case where the gradients are noisy since $y$ cannot be interpolated on the training distribution.
> > > > > > > 2. the case where $y$ is learnable through $\psi$ over the training distribution.

---

> > > > > > > > ### Comment · Reviewer_AoHd · 2021-11-19
> > > > > > > > **Follow up**
> > > > > > > >
> > > > > > > > I understand the difference between these two settings. My point was that if you use the results from the one-pass online setting, then there is no overparameterized model (hence unlearnable). I believe for Figure 5(c) and Figure 5(f), you were arguing about the online setting (in your previous response) with a learnable target function. I think it is wrong.

---

> > > > > > > > > ### Author Response · Authors · 2021-11-19
> > > > > > > > > **Follow up**
> > > > > > > > >
> > > > > > > > > This is the incorrect interpretation of over-parameterized. The setting in Figure 5 has a finite training set ($M=50,000$ MNIST images) which can be fit by the model so it is over-parameterized: the training set is learnable. The reviewer is correct that, because we are operating in the multiple-pass setting, at large time the power law in the test error will not be descriptive since eventually the test error will plateau like the test error in Figure 4 after SGD has passed over the many times. In reality the one-pass theory is a good approximation to the test error in this setting provided that $t m / M$ is not too large. This is the case for example in Figure 5 (f) where $m=5$, $t < 10,000$ and $M=50,000$ which can have at most one pass over the full training set.

---

> > > > > > > > > > ### Comment · Reviewer_AoHd · 2021-11-19
> > > > > > > > > > **Follow up**
> > > > > > > > > >
> > > > > > > > > > Let me say it in a different way.
> > > > > > > > > > 1. For small t, the test curves are similar to the training curves of the ideal world.
> > > > > > > > > > 2. In the ideal world, you have infinite samples. The sample size M = 50,000 is not relevant anymore.
> > > > > > > > > > 3. In the ideal world, I don't know what an overparameterized model means.
> > > > > > > > > >
> > > > > > > > > > I agree that it is always learnable (you could have overparameterized models) in the real world with finite samples, but then you should take the multiple-pass setting (section 4.4). However, I believe the power-law curves are based on the online setting.
> > > > > > > > > >
> > > > > > > > > > Correct me if I'm wrong.

---

> > > > > > > > > > > ### Author Response · Authors · 2021-11-19
> > > > > > > > > > > **Follow up**
> > > > > > > > > > >
> > > > > > > > > > > The last comment is correct in that the theory in section 4.4 should be exact for the setting in Figure 5. The power law theory in Figure 5 is only an approximation for training wide networks, albeit an accurate one. One motivation we had for using this approximation is that it can be easily computed from the kernel gram matrix $K$ (which can be efficiently computed for many architectures in Neural Tangents API)  and the label vector $y$. The theorem for 4.4 requires computing large Fisher information matrices which have size  # of parameters x # of parameters rather than # of data and this operation is not supported as a primitive operation in Neural Tangents.  Further, many prior works have reported power law scalings for wide neural networks [1,2] and our simple power law scalings provide a simple possible explanation for the early time test loss scaling during optimization.
> > > > > > > > > > >
> > > > > > > > > > > [1] https://arxiv.org/abs/1712.00409
> > > > > > > > > > > [2] https://arxiv.org/abs/2102.06701

---

> > > > > ### Author Response · Authors · 2021-11-20
> > > > > **added brief comment in Section 5**
> > > > >
> > > > > To clarify this issue, we added the following sentence in Section 5 when discussing the power law wide network result.
> > > > >
> > > > > "We stress that the exponents $\beta$ were estimated from our one-pass theory, but were utilized experiments on a finite training set. This approximate and convenient version of our theory is quite accurate across these varying models, in line with recent conjectures about early training dynamics (Nakkiran 2021)."

---

### Official Review · Reviewer_My4V · 2021-11-03

**Correctness:** 3
**Technical Novelty And Significance:** 2
**Empirical Novelty And Significance:** 2
**Recommendation:** 5
**Confidence:** 4

**Main Review:**

I find the following pros and cons of the paper.

Pros:

- The paper is written clearly. The problem setup is described rigorously and cleanly, and the content of the paper is well-organized.

- The results of the paper are solid. The theoretical results are correct, and the experiments are also reasonable.

Cons:

- The theoretical results in this paper may be a bit incremental given existing results. This paper misses some important related works on learning overparameterized linear models in linear regression, ridge regression, and logistic regression [2,3,4,5,6]. Notably, all the works above also focus on the relation between the test loss/error and the structure of the data. Therefore the authors should discuss how the results in this paper are different from these existing works.

- This paper claims to cover the training of deep neural networks. This may be overselling. The connection between the analysis in this paper and deep neural networks is mainly through the recent results on Neural Tangent Kernel. However, it is believed in the literature that the NTK setting cannot represent deep neural networks. Moreover, according to [1], neural networks under the NTK setting corresponds to a random feature model, and in literature, the random feature models are commonly studied based on its Gram matrix, just like what has been done in [1,2,3,4,5,6]. However, the results in this paper are all given in terms of the data covariance matrix, which does not seem appropriate for possibly infinite-dimensional feature maps or random feature maps.

- The study in this paper is not very comprehensive either. Although the authors present theories as well as experiments, they are not exactly to back up each other. Instead, it seems that many of the experiments are presented to handle results that cannot be covered by theories, such as the optimal mini-batch size under a fixed compute budget. I think it could greatly improve the quality of theories of the paper if the authors can use them to more comprehensively predict/explain the experiment results.


[1] Cao and Gu. "Generalization Bounds of Stochastic Gradient Descent for Wide and Deep Neural Networks." NeurIPS 2019

[2] Bartlett et al. "Benign overfitting in linear regression.", PNAS, 2020

[3] Tsigler and Bartlett. "Benign overfitting in ridge regression." arXiv:2009.14286, 2020

[4] Chatterji and Long. "Finite-sample analysis of interpolating linear classifiers in the overparameterized regime." JMLR, 2021

[5] Zou et al. "Benign Overfitting of Constant-Stepsize SGD for Linear Regression", COLT 2021

[6] Cao et al. "Risk bounds for over-parameterized maximum margin classification on sub-gaussian mixtures." NeurIPS 2021


**Summary Of The Paper:**

This paper studies the relation between the test error of stochastic gradient descent on training linear models and the structure of the data distribution, the iteration number, and the batch size. The analyses are first done on one-pass SGD and then extended to multi-pass SGD. Results in theory and real data experiments are presented to illustrate the learning curves of the SGD algorithm.

**Summary Of The Review:**

Based on the cons listed in the Main Review section, I would like to give a rating of 5 for the current version of the submission.

---

> ### Author Response · Authors · 2021-11-18
> **Response to My4V**
>
> We thank the reviewer for carefully reading our paper and providing feedback on our related works and the discussion of the power law setting.
>
> 1. We thank the reviewer for bringing the works [1-6] to our attention. First, the papers [1-4] and [6] study the offline generalization performance of fully trained models in the over-parameterized regime. We now cite these when discussing the performance of offline estimators in our related work. The reference in [5] is especially relevant to our work since it analyzes the performance of SGD throughout training and derives a power law $1/t^2$ bound under mild conditions. The regularity conditions on fourth moments used in this work are similar to what we needed in Theorem 4.3.
>
> 2. However, we would again like to stress that our full theory exactly characterizes the average (over history of sampled batches) error of SGD for arbitrary features (in terms of second and fourth moments) and we also provide a simpler second-moment based theory which is exact for Gaussian features. These exact results are novel with this work and show how eigenvalues learning rate and batch size interact during learning. We show that a variety of scalings are possible depending on the precise choice of feature map $\psi(x)$, data distribution, learning task $y(x)$ and hyperparameters.
>
> 3. The reviewer is correct in identifying that our theory only applies to neural networks in the 'lazy' training regime where they can be approximated with the NTK. We added a sentence in the Novel Contributions section which clarifies our NN experiments "This theory is shown to be accurate in experiments with random feature models and wide networks in the kernel regime trained on MNIST and CIFAR-10." We also added a sentence in the conclusion: "While our theory accurately describes networks in the lazy training regime, average case learning curves in the feature learning regime would be interesting future extension." Although our work does not cover deep neural networks in the rich feature learning regime where the features are learned and the kernel evolves, we believe the ability to describe width-500 neural networks with trained on real data (Figure 5 shows real fully trained NNs with NTK parameterization) is still an achievement.
>
> 4. We do not write everything in terms of the data correlations: our theory agrees with the cited works in expressing everything in terms of *feature correlation eigenvalues*, not in terms of data correlation eigenvalues. This is why, for instance, the different feature maps in Figure 5 have different learning curves on the same dataset. We want to stress that the correlation matrix of interest is $\Sigma = \left< \psi(x) \psi(x)^\top \right>_x$.
>
>
> 5. The experiments we perform attempted to demonstrate the validity of our theoretical expressions. The dashed black curves are computed apriori from spectral information and our theory before the experiments (solid color lines) are performed. The good match supports our theory.  Perhaps we misunderstood the question. We would appreciate any more information about how the reviewer thinks our experiments could be improved to support our theory.

---

### Official Review · Reviewer_Ack9 · 2021-11-04

**Correctness:** 4
**Technical Novelty And Significance:** 3
**Empirical Novelty And Significance:** Not applicable
**Recommendation:** 6
**Confidence:** 4

**Main Review:**


I really liked reading this paper. Overall, it is well written and the results are presented in a clear way. It points towards a very important direction of investigation that is attracting increasing interest in machine learning theory. However, the discussion is missing some very relevant references that already cover some of the results, and therefore the authors over-claim some of their contributions.
Missing references:

(i)  Ref. [1] derives rigorous conditions under which learning with data coming from a single-layer generator can be analysed via an equivalent Gaussian model. Moreover, they show for a broader class of data distributions — in particular, pre-trained deep generators — that the theoretical predictions for both online-SGD and full-batch learning match the experimental curves for the loss;

(ii) Ref. [2] extends the above work to include target distributions generated from a pre-trained teacher with a static feature map.

These works also consider other losses than the MSE (e.g., logistic loss). I believe that the authors should clarify the significance of their contributions with respect to (i) and (ii).

The paper presents interesting results on the optimal hyper-parameters, i.e., learning rate and batch size, which could lead to useful guidance for practitioners. In particular, the authors highlight the dependence of the optimal batch size on the power-law decay of the feature distribution and on the noise level of the target distribution.

Minor comment: Figure 2(a) would be more readable if there was a legend indicating the correspondence between the values of $b$ and the different colors.

Figure 3(a): can the authors provide an intuitive explanation of why the mismatch between theory and simulations increases with the learning rate? In particular, why do simulations seem to consistently stay below theoretical curves? On how many seeds have the simulations been averaged over?

In this paper the compute budget $C=tm$ is somewhat assumed to be a good measure of computational limitations, and a tradeoff between performance and compute budget implicitly emerges as a suggested goal. However, gradient evaluations can be efficiently parallelized for large batch sizes, and it is often argued that settings where large batches still generalize well are a desirable goal (see, for instance, [3]). I would like to have some clarifications on this point.

The most interesting and novel aspect of the work in my opinion is the study of multi-pass gradient descent, which — to the best of my knowledge — has only been previously done in [4,5]. The setting in [4,5] encompasses more general sampling procedures and generic (possibly non-convex) loss functions, at the price of a more complicated system of equations for the learning curves. On the contrary, in this paper the final expressions for the test and train losses are nice and simple. It seems to me that equations (71) can only be obtained for sampling with replacement since the weights ${\bf w}(t)$ at time $t$ must be independent on the mini-batch drawn to compute ${\bf w}(t+1)$. I would appreciate if the authors could elaborate more on the limitations/potential extensions of their analysis to more complicated loss functions and different sampling procedures/algorithms (e.g., SGD with momentum, sampling without replacement, curriculum learning…).

Minor comment: I was surprised to see that the scaling of fluctuations turns out to be $\eta^2/m$, at variance with what is usually assumed ($\eta/m$) in works that model the flow dynamics of SGD as a Langevin-like process (see, e.g., [6]). I do not know if this point is interesting/worth of comments.

Some typos are present both in the text and in the equations. For instance:
-  The left-hand side of eq. (4) is missing $y_{\perp}(x)$;
-  Right below eq. (6) the average brackets are missing: $L_t$ should be $\langle L_t \rangle$;
-  The left-hand side of eq. (67), Appendix I should be ${\bf w}_{t+1}$;
-  Appendix I ends with an incomplete sentence.

References:

[1] arXiv:2006.14709

[2] arXiv:2102.08127

[3] arXiv:1811.03600

[4] arXiv:2006.06098

[5] arXiv:2103.04902

[6] arXiv:1711.04623

**Summary Of The Paper:**

This paper addresses the problem of characterising analytically the dynamics of stochastic gradient descent (SGD) in problems where the data have features with arbitrary covariance structure. All the results are valid for linear models (e.g., random features, neural tangent kernel) trained on the mean-squared error loss. First, the authors consider one-pass SGD and Gaussian features, and derive an exact closed-form expression for the time-evolution of the expected test loss, i.e., averaged over the data distribution and the sampling sequence. Then, they turn to generic non-Gaussian feature maps with arbitrary covariance and similarly compute an exact closed-form expression for the expected test loss of one-pass SGD. They also derive an upper bound that only depends on the features covariance, provided that a regularity condition holds on the fourth moments of the features. This result shows that non-Gaussian effects are negligible in the settings under consideration. Finally, they extend their results to multi-pass SGD and provide expressions for the expected test and training losses over time. The authors apply their theoretical findings to derive heuristic estimates of the optimal batch size and learning rate and study their dependence on the features and target distributions. The numerical simulations show a good agreement with the theoretical predictions.

**Summary Of The Review:**

Although the paper is well written and points towards a relevant direction of research, I believe that the contributions are rather incremental with respect to previous literature that is not taken into account. Therefore, I think that at the present stage this work is not ready for publication.

*** Update after reading the authors response***

The authors have carefully addressed my concerns and improved their manuscript accordingly. I recommend the updated version for publication and I have therefore raised my score to 6.

---

> ### Author Response · Authors · 2021-11-18
> **Response to Ack9 Part 1**
>
> We thank the reviewer for their useful comments, bringing our attention to these great related works and encouraging us to state clearly our novel contributions.
>
> 1. We thank the reviewer for requesting a discussion and clarification of these great related works. Based on the comments from a variety of reviewers, we rewrote our related works and listed concretely the novel contributions of our paper. We will now discuss concretely the references this reviewer mentioned. We tried to prevent overclaiming and make clear that we only study MSE loss, though for arbitrary feature structure.
>
> 2. The beautiful work in [1] describes high dimensional vectors generated by a lower dimensional hidden manifold, which leads to correlated input data. In this high ambient dimension limit, the authors characterized the learning dynamics in 2 layer networks in continuous time for a variety of nonlinearities. The model the authors used for the data is $X=g(C F / \sqrt{D})$ for $C \in \mathbb{R}^{P \times D}$ and $F \in \mathbb{R}^{D \times N}$ where $g$ is a nonlinear function. Our paper does not require assuming such a form for the data, but rather computes everything in terms of moments of the induced features $\psi(x)$. Also unlike our analysis, the resulting loss curves in this theory concentrate and do not exhibit stochastic fluctuations due to the stochastic estimation of gradients. Our work, on the other hand, studies SGD in discrete time so fluctuations in gradients have significant impact on dynamics, showing how learning rate, batch size, and data structure interact to alter the learning curves. This allows us to study computational tradeoffs in batch size.
>
> 3. The work in [2] focuses on the offline learning curves of generalized linear models with structured data for a variety of loss functions (not just MSE). However, since we study the online loss during optimization, the interpretations of our theories are different. Specifically, the theory in [2] estimates asymptotes of our plots in Figure 4 (b). For instance, the theory in [2] is agnostic about optimization procedure and hyperparameters $(\eta,m)$ but considers the generalization of the training loss minimizer. We added this citation to the Related work when discussing offline generalization error.
>
>
> 4. We added a colorbar for the different values of $b$ in Figure 4.
>
>
> 5. In Figure 3(d,e): we did not average over many random trajectories but showed only one trajectory. We reran the experiment with averaging over 20 trials which is the new Figure 3(d,e,f). This appears to account for the discrepancy between the large learning rate experiment and theory, though a discrepancy still remains on the large noise experiment in f.
>
>
> 6. The paper in [3] is very interesting since it discusses data parallelism which is achievable with modern hardware. In this setting where $m_{parallel}$ gradients can be simultaneously computed, the training time cost may not scale with $C = t m$ but rather with $t$ directly provided that $m \leq m_{parallel}$. Under this condition the fastest wall-clock time is achieved only for $m\geq m_{parallel}$, since, as we establish in the Appendix, increasing $m$ at fixed $t$ always reduces expected test loss. We added a comment about this work in section 4.1. We added a sentence about the limitations of studying fixed $C$ learning curves based on this observation. We think that $C$ is still meaningful in the one-pass setting where $C$ represents the sample complexity of the algorithm even when it does not represent total run time.

---

> ### Author Response · Authors · 2021-11-18
> **Response to Ack9 Part 2**
>
> 7. We thank the reviewer for appreciating the multiple pass setting portion of our paper. Indeed, we need each individual sample to be drawn from the dataset independently which requires sampling with replacement. Sampling without replacement or in a fixed curriculum (such as cycles over the train set in steps of size $m$) alters the distribution of features at each timestep.
>
> 8. On the question of the limitations of our approach and possible future extensions, we have a few comments. First, the current theory is simple since it requires computing spectral statistics of both training and test distributions separately. Part of the complexity in papers [4,5] arises from averaging the asymptotic error over the *full random training sets* of size $M$ as well as over the random minibatches. In our approach, by contrast, we average over minibatch history for a fixed training set, allowing us to deal with structured data. We think an interesting extension of our work could be to calculate average SGD loss curves over *both* the distribution of random training sets and over the minibatches of size $m$ for structured data in discrete time.
>
> 9. We thank the reviewer for taking an interest in possible extensions of our analysis. We are currently exploring calculation of learning curves for SGD with momentum and curriculum learning setting. We have some preliminary results along these lines but believe they may be beyond the scope of the current draft. For curriculum learning we believe modeling a time dependent data distribution $p_t(x)$ would allow one to compute average loss curves. For momentum, the equations for expected loss close in terms of 3 matrices rather than the single matrix $C_t$ studied in this problem. Though we have not come up with a way of dealing with sampling with replacement, we would be glad if such an extension can be discovered. We added a note in the conclusion that we plan extend our framework to these settings. Other loss functions would also be interesting, but for general $\ell(w \cdot \psi)$, higher moments may be necessary to characterize learning curves for non-Gaussian features.
>
>
>
> 11. As to the scaling with $\eta^2 / m$, this arises from the variance of the update $g_t = \frac{\eta}{m} \sum_{\mu} \psi_\mu \psi_\mu^\top \Delta_t$ conditional on $\Delta_t$, which has covariance matrix
>
> $$ \left< g_t(\psi) g_t(\psi)^\top \right>_{\psi} - \left< g_t(\psi) \right> \left<g_t(\psi)^\top \right> = \frac{\eta^2}{m} [\left< \psi\psi^\top \Delta_t \Delta_t^\top \psi\psi^\top \right> - \Sigma \Delta_t \Delta_t^\top \Sigma]$$
>
> This is not at odds with the update in [6] which uses $w_{t+1} = w_t - \eta \nabla L(w) + \frac{\eta}{\sqrt{m}} \epsilon_t$. Calculating the variance of the update $\nabla L(w) + \frac{\eta}{\sqrt{m}}\epsilon_t$ again gives something $O(\frac{\eta^2}{m})$.
>
> One nice thing about our calculation is that it exactly describes how the noise $\epsilon_t$ evolves during gradient descent. For Gaussian features
> \begin{align}
> \left< \epsilon_t \epsilon_t^\top \right> = \Sigma C_t \Sigma + \Sigma \text{Tr} \Sigma C_t = \Sigma C_t \Sigma + \Sigma \mathcal L_t
> \end{align}
> This shows how the scale of the noise decreases throughout SGD which is why achieving zero error is possible as $t\to\infty$.
>
> 12. Minor issues: Thank you for pointing these out. We addressed them in the newer draft.

---

> > ### Comment · Reviewer_Ack9 · 2021-11-28
> > **Response to authors**
> >
> > I thank the authors for carefully addressing my concerns and for providing some useful clarifications. I have appreciated the effort in improving the manuscript, that is now more solid and, in my opinion, worth publication.

---

### Official Review · Reviewer_FVgD · 2021-11-06

**Correctness:** 3
**Technical Novelty And Significance:** 3
**Empirical Novelty And Significance:** 3
**Recommendation:** 8
**Confidence:** 3

**Main Review:**


The paper is generally well written, although exposition in the main text or in the appendix could be made a bit more pedagogical.
The results it provides are very interesting. The implication of the main theorems on batch sizes and learning rates and the connection with neural tangent kernel theory are carefully explored. Numerical experiments are extensive and well thought.


Some additional comments:

- Use \lVert \lambda \rVert  instead of |\lambda| at the top of pag. 4 for consistency.

- Could discuss the relation with the hidden manifold model where a linear teacher acts on the hidden space, y(x) = w*\cdot x, if the authors think there is anything interesting to
report.

- Theorem 3.2: make explicit t depence  in c_k
- Theorem 3.4: this is assuming Gaussian features as in Theorem 3.1, right? If so, should be clearly stated.

- Theorem 3.5: Why these expressions don't involve 4-th moments? Is some Gaussianity assumption made also here?
- Theorem 3.5: make t dependence explicit in Ltrain and Ltest
- In the MNIST and CIFAR tasks, the regression targets are 0s and 1s?
 - Related Work: "Their predicted exponent
in the small learning rate limit agrees with the exponent we derive with the saddle point approximation in Section 3.1.2.". I find no mention in the main nor the appendix of
saddle point approximations.

- Appendix A: "Since the basis of kernel eigenfunctions
(including the zero eigenvalue functions) is complete over the space of square integrable functions." should end with a comma instead?
- Appendix A: the first N eigenvalues of the kernel K are the same as the eigenvalues of N? The connection could be made more clear and in general this appendix expanded a bit to be more pedagogical.


**Summary Of The Paper:**

The authors derive exact and non-asymptotic (in size or time) expressions for the expected test loss of a linear model during the SGD training dynamics.
The time-dependent average test loss is given as function of the eigenvectors and the eigenvalues of the covariance matrix of the features when the features are Gaussian.
For non-Gaussian features also forth-moments are involved and the Gaussian formula can be adapted to provide an upper bound. In any case, the formula for the
Gaussian case seems to match very well the experiments on real world datasets.
The formalism covers both the online setting and the setting of multiple passes over a fixed training set.

**Summary Of The Review:**

The paper is a nice contribution to the theory of SGD in the quite fundamental linear regression setting. It is well written, contains original and relevant results, and provides good experimental verification. Therefore I recommend acceptance on ICLR without esitation.

---

> ### Author Response · Authors · 2021-11-18
> **Response to FVgD**
>
> We thank the reviewer for carefully reading the paper and providing useful feedback. We tried implemented the suggested changes which we discuss below.
>
> 1. We fixed page 4
>
> 2. We added a longer comparison to the hidden manifold model works in the Related works section. The hidden manifold model of data stipulates a generative expression for the data $X=g(C F / \sqrt{D})$ for $C \in \mathbb{R}^{P \times D}$ and $F \in \mathbb{R}^{D \times N}$ where $g$ is a nonlinear function, $F$ are fixed features of dimension $N$ and $C$ are latent variables of dimension $D$. We instead compute everything in terms of induced moments of the features such as $\left< \psi_i(x) \psi_j(x) \right>_{x }$ and $\left< \psi_i(x) \psi_j(x) \psi_k(x) \psi_\ell(x) \right>$ (which does not require stipulating a generative model for the data $X$) . An additional difference is that our learning curves are in discrete time where the loss and gradients can exhibit fluctuations due to random sampling whereas the learning curves in the hidden manifold model are in continuous time and correspond to gradient flow over a set of order parameters. The stochasticity in our setting can have a nontrivial influence on loss dynamics in discrete time.
>
> 3. We expressed the explicit $t$ dependence on $c_k$, thanks for pointing this out.
>
>
> 4. We fixed the $\lambda$ at the top of page 4.
>
>
> 5. Theorem 3.4: Yes this is in the Gaussian setting. We made this clear in the revised draft. A non-Gaussian version of this result can also be obtained but the formula is more complicated.
>
> 6. Theorem 3.5: In the first submitted draft, this expression is also using a Gaussianity assumption. We rewrote this so that it reflects the general case. In general, for the training distribution correlation $\Sigma$ and fourth moment $\left< \psi_i \psi_j \psi_j \psi_k \right>$ over the training measure, we have
>
> $$ C_t = \left( I - \eta \Sigma \right) C_t (I - \eta \Sigma) + \frac{\eta^2}{m}\left< \psi \psi^\top C_t \psi \psi^\top \right> - \frac{\eta^2}{m} \Sigma C_t \Sigma$$
>
> However, for simplicity, we merely compute the Gaussian theory for the plot in Figure 4. We added the explicit $t$ dependence in $L_{train}$ and $L_{test}$
>
> 7. For the MNIST/CIFAR experiments involving two classes (Figures 2, etc) we use targets in $y \in \{\pm 1\}$, while for the multi-class setting we use one-hot labels as the reviewer asks. We provided more information about the labeling in the Appendix J.
>
>
> 8. Saddle Point: We use a saddle point approximation in Appendix G to obtain the large $t$ scaling in the appendix section on power laws. We added a reference to Appendix G and its integral approximation in the main text section 4.1.2.
>
> 9. Appendix A: we implemented the suggested changes in the appendix and showed the link between the spectra $\Sigma$ and $K(x,x')$.

---

### Author Response · Authors · 2021-11-18
**Comments to all reviewers**

Again, we would like to thank all reviewers for their careful reading and useful suggestions. There were a few repeating comments from the reviewers which we hope to address globally as well as in our individual responses. These concerns were
1. A need to more clearly state our novel contributions. To address this, we added bullet points at the top of page 2 listing our main novel contributions: an exact solution to SGD on MSE loss in terms of 2nd and 4th moments, an exact theory for Gaussian features, study of scalings in special cases, analysis of optimal hyperparameters (see for instance new version of Figure 2), experimental verification of our theory on random feature models and wide NNs on real data, and an extension of our theory to multiple pass SGD on fixed finite training set.
2. Related works came at the end and missed some important prior works. To address this, we incorporated the suggested related works and expanded our discussion of them. We additionally moved the related works section to page 2 so that it comes before our main results.
3. A request to further explain/illustrate some of the optimal hyperparameters results. We added Figure 2 b,c. In 2b show how optimal batch size scales with feature eigenvalues and 2c shows an example of theoretical loss at fixed compute in the $(\eta,m)$ plane.
4. Figure 3 (d,e,f) seemed to have mismatch between experiment and theory. To address this, we ran the experiment over 20 experimental SGD trials and averaged (d and e were not averaged before). The agreement in the loss curves is much better.

---

### Decision · Program_Chairs · 2022-01-20

**Decision:**

Accept (Poster)

**Comment:**

The work presented in this study gives a theoretical finite-sample generalisation performance of stochastic gradient descent on linear models, for different batch-sizes and feature structures. This approach enable the authors to predict the training and test losses of neural networks on real data.

While there were some parts that were initially mis-understood by some reviewers in the initial version of the papers, the extensive discussions between the authors and the reviewers led to several updates, both in the reference to prior work, but also in the presentation clarity. The wide impact and relevance to ICLR of this type of contribution made us recommend this work for acceptance at ICLR.